# Bucks for Buckets (B4B): Active Defenses Against Stealing Encoders

**Jan Dubiński** [1,2,*,†]   **Stanisław Pawlak** [1,†]   **Franziska Boenisch** [4,†]
**Tomasz Trzciński** [1,2,3]   **Adam Dziedzic** [4,*,‡]
[1]Warsaw University of Technology   [2]IDEAS NCBR   [3]Tooploox
[4]CISPA Helmholtz Center for Information Security

## Abstract

Machine Learning as a Service (MLaaS) APIs provide ready-to-use and high-utility encoders that generate vector representations for given inputs. Since these encoders are very costly to train, they become lucrative targets for model stealing attacks during which an adversary leverages query access to the API to replicate the encoder locally at a fraction of the original training costs. We propose *Bucks for Buckets (B4B)*, the first *active defense* that prevents stealing while the attack is happening without degrading representation quality for legitimate API users. Our defense relies on the observation that the representations returned to adversaries who try to steal the encoder's functionality cover a significantly larger fraction of the embedding space than representations of legitimate users who utilize the encoder to solve a particular downstream task. B4B leverages this to adaptively adjust the utility of the returned representations according to a user's coverage of the embedding space. To prevent adaptive adversaries from eluding our defense by simply creating multiple user accounts (sybils), B4B also individually transforms each user's representations. This prevents the adversary from directly aggregating representations over multiple accounts to create their stolen encoder copy. Our active defense opens a new path towards securely sharing and democratizing encoders over public APIs.[5]

## 1   Introduction

In model stealing attacks, adversaries extract a machine learning model exposed via a public API by repeatedly querying it and updating their own stolen copy based on the obtained responses. Model stealing was shown to be one of the main threats to the security of machine learning models in practice [38]. Also in research, since the introduction of the first extraction attack against classifiers [40], a lot of work on improving stealing [27, 33, 40, 41], extending it to different model types [8, 37], and proposing adequate defenses [18, 25, 26, 31] has been put forward. With the recent shift in learning paradigms from supervised to self supervised learning (SSL), especially the need for new defenses becomes increasingly pressing. From an academic viewpoint, the urge arises because it was shown that SSL models (*encoders*) are even more vulnerable to model stealing [16, 29, 36] than their supervised counterparts. This is because whereas supervised models' output is low dimensional, *e.g.,* per-class probabilities or pure labels, SSL encoders output high-dimensional representation vectors that encode a larger amount of information and thereby facilitate stealing. In addition, from a practical industry's viewpoint, defenses are required since many popular API providers, such as Cohere, OpenAI, or Clarify [1–3] already expose their high-value SSL encoders via APIs to a broad range of users.

---

[*]Corresponding authors: jan.dubinski.dokt@pw.edu.pl and adam.dziedzic@cispa.de

[†]Equal contribution.

[‡]Project Lead.

[5]Code available at `https://github.com/stapaw/b4b-active-encoder-defense`

37th Conference on Neural Information Processing Systems (NeurIPS 2023).

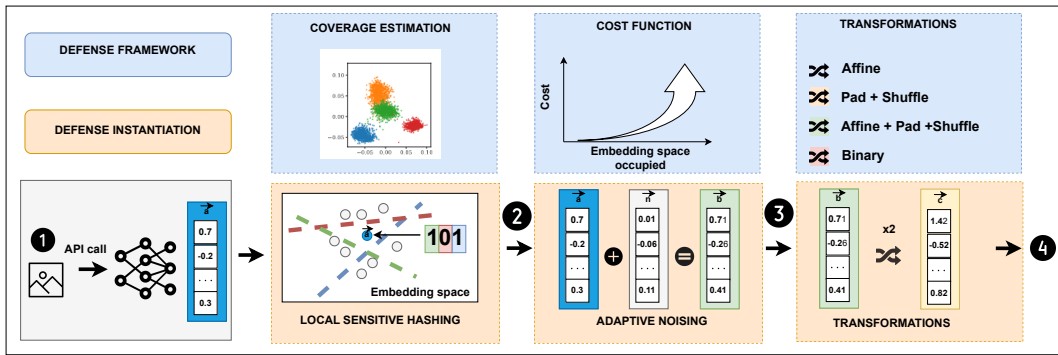

Figure 1: **Overview of B4B.** In the upper part, we present our B4B framework that consists of three modular building blocks: (1) A coverage estimation to track the fraction of embedding space covered by the representations returned to each user, (2) a cost function that serves to map the coverage to a concrete penalty to prevent stealing, and (3) per-user transformations that are applied to the returned representations to prevent sybil attacks. In the lower part, we present a concrete instantiation of B4B and the operation flow of our defense: ❶ The API calculates representations for the incoming queries. ❷ We instantiate the coverage estimation with local sensitive hashing and estimate the covered space as the fraction of *hash buckets* occupied. We calibrate the costs by adding noise to the representations according to the coverage. ❸ We apply a set of transformations on a per-user basis. ❹ The noised and transformed representations are returned to the user.

Most of the current defenses against encoder stealing are *reactive*, *i.e.,* they do not actively prevent the stealing but rather aim at detecting it by adding watermarks to the encoder [14, 16] or performing dataset inference to identify stolen copies [17]. Since at the point of detection, the damage of stealing has already been inflicted, we argue that reactive defenses intervene too late and we advocate for *active* defenses that prevent stealing while it is happening. Yet, active defenses are challenging to implement because they not only need to prevent stealing but also should preserve the utility of representations for legitimate users. The only existing active defense against encoder stealing [29] falls short on this latter aspect since it significantly degrades the quality of representations for all users.

To close the gap between required and existing defenses, we propose *Bucks for Buckets (B4B)*, the first active defense against encoder stealing that does not harm utility for legitimate users. B4B leverages the observation that the representations returned to adversaries who try to steal the encoder's functionality cover a significantly larger fraction of the full embedding space than representations of legitimate users who utilize the encoder to solve a particular downstream task. To turn this observation into a practical defense, B4B is equipped with three modular building blocks: (1) The first building block is a tracking mechanism that continuously estimates the fraction of the embedding space covered by the representations returned to each user. The intuition why this is relevant is that by covering large fractions of the embedding space, the representations will suffice for an adversary to reproduce the encoder's functionality, *i.e.,* to successfully steal it. (2) B4B's second building block consists of a cost function to translate the covered fraction of the embedding space into a concrete penalty. We require this cost function to significantly penalize adversaries trying to steal the model while having only a minimal effect on legitimate users. (3) The third building block contains transformations that can be applied to the representations on a per-user basis to prevent adaptive attackers from circumventing our defense by creating multiple user accounts (sybils) and distributing their queries over these accounts such that they minimize the overall cost. We present the different building blocks of B4B in Figure 1.

While B4B's modularity enables different instantiations of the three building blocks, we propose a concrete end-to-end instantiation to showcase the practicability of our approach. To implement tracking of the covered embedding space, we employ *local sensitive hashing* that maps any representation returned to a given user into a set of hash **buckets**. We base our cost function (*i.e.,* the ***"bucks"***) on utility and make B4B add noise to the representations with a magnitude that increases with the number of buckets occupied by the given user. While the scale of noise added to legitimate users'

representations does not harm their downstream performance due to their small embedding space coverage, the representations returned to an adversary become increasingly noisy—significantly degrading the performance of their stolen encoder. Finally, we rely on a set of transformations (*e.g.,* affine transformations, shuffling, padding) that preserve downstream utility [17]. While, as a consequence, legitimate users remain unaffected by these transformations, adversaries cannot directly combine the representations obtained through different sybil accounts anymore to train their stolen copy of the encoder. Instead, they first have to remap all representations into the same embedding space, which we show causes both query and computation overhead and still reduces the performance of the stolen encoder.

In summary, we make the following contributions:

1. We present B4B, the first active defense against encoder stealing that does not harm legitimate users' downstream performance. B4B's three building blocks enable penalizing adversaries whose returned representations cover large fractions of the embedding space and prevent sybil attacks.

2. We propose a concrete instantiation of B4B that relies on local sensitive hashing and decreases the quality of representations returned to a user once their representations fill too many hash buckets.

3. We provide an end-to-end evaluation of our defense to highlight its effectiveness in offering high utility representations for legitimate users and degrading the performance of stolen encoders in both the single and the sybil-accounts setup.

## 2   Related Work

**Model Extraction Attacks.**   The goal of the model extraction attacks is to replicate the functionality of a victim model $f_v$ trained on a dataset $D_v$. An attacker has a black box access to the victim model and uses a stealing dataset $D_s = \{q_i, f_v(q_i)\}_{i=1}^n$, consisting of queries $q_i$ and the corresponding outputs $f_v(q_i)$ returned by the victim model, to train a stolen model $f_s$. Model extraction attacks have been shown against various types of models including classifiers [24, 40] and encoders [16, 36].

**Self Supervised Learning and Encoders.**   SSL is an increasingly popular machine learning paradigm. It trains encoder models to generate representations from complex inputs without relying on explicit labels. These representations encode useful features of a given input, enabling efficient learning for multiple downstream tasks. Many SSL frameworks have been proposed [9, 10, 12, 22, 23, 44]. In our work, we focus on the two popular SSL vision encoders, namely SimSiam [12] and DINO [9], which return high-quality representations that achieve state-of-the-art performance on downstream tasks when assessed by training a linear classifier directly on representations. SimSiam trains with two Siamese encoders with directly shared weights. A prediction MLP head is applied to one of the encoders $f_1$, and the other encoder $f_2$ has a stop-gradient, where both operations are used for avoiding collapsing solutions. In contrast, DINO shares only architecture (not weights) between a student $f_1$ and a teacher model $f_2$, also with the stop-gradient operation, but not the prediction head. While SimSiam uses convolutional neural networks (CNNs), DINO also employs vision transformers (ViTs). Both frameworks use a symmetrized loss of the form $\frac{1}{2}g(f_1(x_1), f_2(x_2)) + \frac{1}{2}g(f_1(x_2), f_2(x_1))$ in their optimization objectives, where $g(\cdot, \cdot)$ is negative cosine similarity for SimSiam and cross-entropy for DINO. SimSiam and DINO's similarities and differences demonstrate our method's broad applicability across SSL frameworks. More details can be found in Appendix E.

**Stealing Encoders.**   The stealing of SSL encoders was shown to be extremely effective [16, 29, 36]. The goal of extracting encoders is to maximize the similarity of the outputs from the stolen local copy and the original representations output by the victim encoder. Therefore, while training the stolen copy, the adversary either imitates a self-supervised training using a contrastive loss function, *e.g.,* InfoNCE [10] or SoftNN [21] or directly matches both models' representations via the Mean Squared Error (MSE) loss. To reduce the number of queries sent to the victim encoder, the attack proposed in [29] leverages the key observation that the victim encoder returns similar representations for any image and its augmented versions. Therefore, a given image can be sent to the victim while the stolen copy is trained using many augmentations of this image, where the representation of a given augmented image is approximated as the one of the original image produced by the victim encoder.

**Defending Encoders.**   Recently, watermarking [7, 25, 42] methods have been proposed to detect stolen encoders [14, 16, 43]. Many of these approaches use downstream tasks to check if a watermark embedded into a victim encoder is present in a suspect encoder. Dataset inference [30] is another type

of encoder ownership resolution. It uses the victim's training dataset as a unique signature, leveraging the following observation: for a victim encoder trained on its private data as well as for its stolen copies, the distribution of the representations generated from the victim's training data differs from the distribution of the representations generated on the test data. In contrast, for an independently trained encoder, these two distributions cannot be distinguished, allowing the detection of stolen copies [17]. However, all the previous methods are *reactive* and aim at detecting the stolen encoder instead of *actively* preventing the attack. The only preliminary active defenses for encoders were proposed by [16, 29]. They either perturb or truncate the answers to poison the training objective of an attacker. These operations were shown to harm substantially the performance of legitimate users, which renders the defense impractical. In contrast, our B4B has negligible impact on the quality of representations returned to legitimate users.

## 3    Actively Defending against Model Stealing with B4B

B4B aims at actively preventing model stealing while preserving high-utility representations for legitimate users. Before introducing the three main building blocks of B4B, namely (1) the estimation of embedding space coverage, (2) the cost function, and (3) the transformation of representations (see Figure 1), we detail our threat model and the observation on embedding space coverage that represents the intuition behind our approach.

### 3.1    Threat Model and Intuition

Our setup and the resulting threat model are inspired by public APIs, such as Cohere, OpenAI, or Clarify [1–3] that expose encoders to users through a pre-defined interface. These encoders are trained using SSL on large amounts of unlabeled data, often crawled from the internet, and therefore from diverse distributions. We notice that to provide rich representations to multiple users, the training dataset of the encoder needs to be significantly more diverse than the individual downstream tasks that the users query for representations. For instance, if the encoder behind the API is trained on the ImageNet dataset, then the legitimate users are expected to query the API for downstream tasks, such as CIFAR10 or SVHN. Similarly, if the encoder is trained on CIFAR10, the expected downstream tasks are MNIST or Fashion MNIST. Yet, in the design of our defense, we consider adversaries who can query the encoder with arbitrary inputs to obtain high-dimensional representation vectors from the encoder. Our defense is independent of the protected encoder's architecture and does not rely on any assumption about the adversary's data and query strategy.

We argue that even in this restricted setup, our defense can distinguish between adversaries and legitimate users by analyzing the distribution of representations returned to them. In Figure 2, by using PCA to project representations for different datasets to a two-dimensional space, we visualize that representations for different downstream tasks cluster in *disjoint* and *small sub-spaces* of the full embedding space. The representations were obtained from a SimSiam encoder originally trained on ImageNet (we observe similar clustering for DINO shown in Appendix F). As a result, legitimate users can be characterized by their representations' small coverage of the embedding space. In contrast, the adversary does not aim at solving a particular downstream task. They instead would want to obtain representations that cover large fractions of the embedding space. This enables reproducing the overall functionality of the encoder (instead of only learning some local task-specific behavior). Indeed, it has been empirically shown by prior work, such as [16], that stealing with multiple

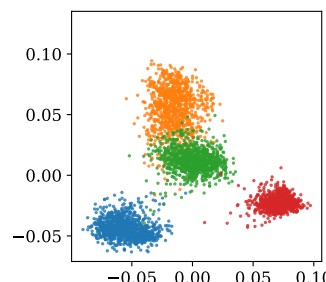

Figure 2: **Representations from Different Tasks Occupy Different Sub-Spaces of the Embedding Space. Presented for Fashion-MNIST, SVHN, CIFAR10, and STL10.**

distributions, *e.g.,* by relying on the complex ImageNet dataset, yields higher performance of the stolen encoder on various downstream tasks than stealing with a downstream dataset, such as CIFAR10. As a result, intuitively, we can identify and penalize adversaries based on their coverage of the embedding space, which will be significantly larger than the coverage of legitimate users. We leverage this intuition to build our B4B defense and present our three main building blocks in the following sections.

## 3.2 Building Block 1: Coverage Estimation of the Embedding Space

The first building block of our B4B serves to estimate and continuously keep track of the fraction of the embedding space occupied by any given user. Let $\mathcal{E}$ denote our embedding space of dimension $s$, further, let $U$ be a user with a query dataset $D = q_1, \ldots, q_n \in \mathcal{D}$ and let $f_v : \mathcal{D} \to \mathcal{E}$ be our protected victim encoder that maps data points from the input to the embedding space. Assume user $U$ has, so far, queried a subset of their data points $q_1, \ldots, q_j$ with $j \leq n$ to the encoder and obtained the representations $r_1, \ldots, r_j$ with each $r_i \in \mathbb{R}^s$. We aim to estimate the true fraction of the embedding space $\mathcal{E}_f^U$ that is covered by all returned representations $r_1, \ldots, r_j$ to user $U$ and denote our estimate by $\tilde{\mathcal{E}}_f^U$.

**Local Sensitive Hashing.**   One of the methods to approximate the occupied space by representations returned to a given user is via Local Sensitive Hashing (LSH) [39]. We rely on this approach for the concrete instantiation of our B4B and use it to track per-user coverage of the embedding space. Standard (cryptographic) hash functions are characterized by high dispersion such that hash collisions are minimized. In contrast, LSH hashes similar data points into the same or proximal, so-called *hash buckets*. This functionality is desired when dealing with searches in high-dimensional spaces or with a large number of data points. Formally, an LSH function $\mathcal{H}$ is defined for a metric space $\mathcal{M} = (M, d)$, where $d$ is a distance metric in space $M$, with a given threshold $T > 0$, approximation factors $f > 1$, and probabilities $P_1$ and $P_2$, where $P_1 \gg P_2$. $\mathcal{H}$ maps elements of the metric space to buckets $b \in B$ and satisfies the following conditions for any two points $q_1, q_2 \in M$: (1) If $d(q_1, q_2) \leq T$, then $\mathcal{H}(q_1) = \mathcal{H}(q_2)$ (*i.e.*, $q_1$ and $q_2$ collide in the same bucket $b$) with probability at least $P_1$. (2) If $d(q_1, q_2) \geq fT$, then $\mathcal{H}(q_1) = \mathcal{H}(q_2)$ with probability at most $P_2$.

## 3.3 Building Block 2: Cost Function Design

Once we can estimate the coverage of an embedding space for a given user $U$ as $\tilde{\mathcal{E}}_f^U$, we need to design a cost function $\mathcal{C} : \mathbb{R}^+ \to \mathbb{R}^+$ that maps from the estimated coverage to a cost. The cost function needs to be designed such that it does not significantly penalize legitimate users while imposing a severe penalty on adversaries to effectively prevent the encoder from being stolen. The semantics of the cost function's range depend on the type of costs that the defender wants to enforce. We discuss a broad range of options in Appendix C. These include monetary cost functions to adaptively charge users on a batch-query basis depending on their current coverage, costs in the form of additional computation that users need to perform in order to obtain their representations, similar to the proof of work in [18], costs in the form of delay in the interaction with the encoder behind the API [4], or costs in form of disk space that needs to be reserved by the user (similar to a proof of space [19, 20]). Which type of cost function is most adequate depends on the defender's objective and setup.

**Exponential Cost Functions to Adjust Utility of Representations.**   In the concrete instantiation of B4B that we present in this work, we rely on costs in the form of the utility of the returned representations. We choose this concrete instantiation because it is intuitive, effective, and can be directly experimentally assessed. Moreover, it is even suitable for public APIs where, for example, no monetary costs are applicable. We adjust utility by adding Gaussian noise with different standard deviation $\sigma$ to the returned representations. Since we do not want to penalize legitimate users with small coverage but make stealing for adversaries with growing coverage increasingly prohibitive, we instantiate an exponential cost function that maps from the fraction of hash buckets occupied by the user to a value for $\sigma$. We choose the general form of this function as

$$f_{\lambda,\alpha,\beta}(\tilde{\mathcal{E}}_f^U) = \lambda \times (\exp^{\ln \frac{\alpha}{\lambda} \times \tilde{\mathcal{E}}_f^U \times \beta^{-1}} - 1) \tag{1}$$

where $\lambda < 1$ compresses the curve of the function to obtain low function values for small fractions of occupied buckets, and then we set a target penalty $\alpha$ for our cost function at a specified fraction of filled buckets $\beta$. For instance, if we want to enforce a $\sigma$ of 5 at 90% of filled buckets (*i.e.,* for $\tilde{\mathcal{E}}_f^U = 0.9$), we would need to set $\alpha = 5$ and $\beta = 0.9$.

## 3.4 Building Block 3: Per-User Representation Transformations against Sybil Attacks

Given that our defense discourages users from querying with a wide variety of data points from different distributions, an adversary could create multiple fake user accounts (sybils) and query

different data subsets with more uniform representations from each of these accounts. By combining all the obtained representations and using them to train a stolen copy, the adversary could overcome the increased costs of stealing. To defend against such sybil attacks, we propose individually transforming the representations on a per-user level before returning them. As a result, the adversary would first have to map all the representations to one single unified space before being able to jointly leverage the representations from different accounts for their stolen copy.

Formally, for a given query $q_i$, the protected victim encoder produces a representation $r_i = f_v(q_i)$, which is transformed by a user-specific transformation $T_U(r_i)$ before being returned to the querying user $U$. For a new user $U$, the defense randomly selects the transformation $T_U$ from all possible choices. Note that the randomness is also added on a per-transformation basis, instead of only on the level of selecting the transformations. For example, a permutation of the elements in the output representations should be different for each user.

We formulate two concrete requirements for the transformations. First, they should preserve utility for legitimate users on their downstream tasks, and second, they should be costly to reverse for an adversary.

**Utility Preserving Transformations.** As a concrete instantiation for our B4B, we propose a set of transformations that fulfill the above-mentioned two requirements: (1) *Affine* where we apply affine transformations to representations, (2) *Pad* where we pad representations with constant values, (3) *Add* where we add constant values at random positions within representations, (4) *Shuffle* where we shuffle the elements in the representation vectors, and (5) *Binary* where the original representations are mapped to binary vectors relying on a random partitioning of the representation space. To preserve the full amount of information contained in the original representations, in our binary transformations, we tune the length of binary representations. We visualize the operation of each of these transformations in Appendix C. All these transformations can additionally be combined with each other, which further increases the possible set of transformations applied per user. This renders it impossible for an adversary to correctly guess and reverse the applied representation. Instead, the adversary has to remap the representations over all accounts into a single embedding space in order to unify them and leverage them for training of their stolen encoder copy. We present an exhaustive list of strategies that adversaries can apply for the remapping in Appendix D. All the attack methods reduce to the minimum of remapping between representations of a pair of users, *i.e.,* they are at least as complex as mapping between two separate accounts. In the next section, we show that our defense already impedes stealing for an adversary with two accounts.

# 4  Empirical Evaluation

We first empirically evaluate our instantiation of B4B's three building blocks and show how to calibrate each of them for our defense. Finally, we provide an end-to-end evaluation that highlights B4B's effectiveness in preserving downstream utility for legitimate users while successfully preventing the stealing by adversaries.

**Experimental Setup.** We conduct experiments on various kinds of downstream tasks and two popular SSL encoders. To test our defense, we use FashionMNIST, SVHN, STL10, and CIFAR10 as our downstream datasets, each with standard train and test splits. For stealing, we utilize training data from ImageNet and LAION-5B. We rely on encoder models from the SimSiam [12] and the DINO [9] SSL frameworks. As our victim encoders, we use the publicly available ResNet50 model from SimSiam trained for 100 epochs on ImageNet and the ViT Small DINO encoder trained for 800 epochs on ImageNet, each using batch size 256. The ViT architecture takes as input a grid of non-overlapping contiguous image patches of resolution $NxN$. In this paper, we typically use $N = 16$. The Simsiam encoder has an output representation dimension of 2048, while DINO returns 1536 dimensional representations. We examine the utility of downstream classifiers using SimSiam's or DINO's representations obtained for the respective downstream datasets. To implement LSH, we rely on random projections [34] that we implement from scratch. For a detailed description of our stealing and downstream training parameters, we refer to Appendix F.

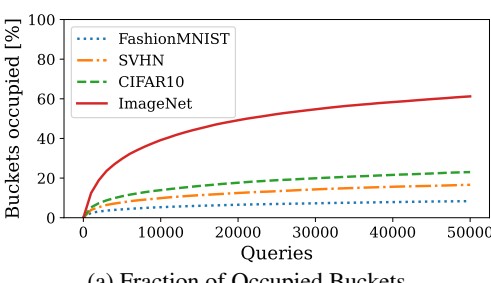 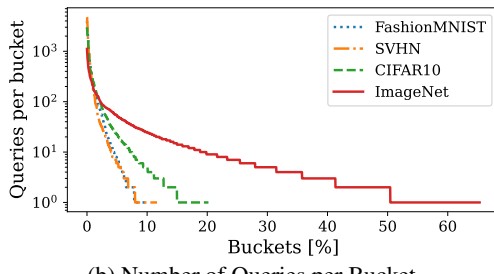

|(a) Fraction of Occupied Buckets. | (b) Number of Queries per Bucket.|

Figure 3: **Estimating Embedding Space Coverage through LSH on SimSiam Encoder.** We present the fraction of buckets occupied by representations of different datasets as a function of the number of queries posed to the encoder *(left)*. We observe that representations for the downstream datasets (FashionMNIST, SVHN, CIFAR10) occupy a smaller fraction of buckets than representations from the complex ImageNet dataset. Our evaluation of the number of queries whose representations are mapped to the same bucket *(right)* indicates that our total number of buckets ($2^{12}$) is well calibrated for the estimation of covered representation space: over all datasets, we experience hash collisions, *i.e.,* queries whose representations are mapped to the same buckets. This indicates that our LSH is capable of representing similarities in the representations.

## 4.1 Local Sensitive Hashing for Coverage Estimation

We first observe that the choice of the total number of hash buckets in the LSH influences the effectiveness of our method. In the extreme, if we have a too large number of buckets, the number of buckets filled will correspond to the number of queries posed by a user which fails to capture that similar representations cover similar sub-spaces of the embedding space, and hence does not serve to approximate the total fraction of the embedding space covered. However, if we have too few buckets, even the representations for simple downstream tasks will fill large fractions of buckets, making it impossible to calibrate the cost function such that it only penalizes adversaries. We experimentally find that for our evaluated encoders, $2^{12}$ buckets represent a good trade-off. In Appendix F.6, we present an ablation study on the effect of the number of total buckets.

Our evaluation of the LSH to track coverage of the embedding space is presented in Figure 3. We observe that representations returned for standard downstream tasks (FashionMNIST, SVHN, CIFAR10) occupy a significantly smaller fraction of the total number of buckets than complex data from multiple distributions (ImageNet, LAION-5B). We present additional experimental results on measuring the coverage of the representation space in Appendix F.5. Specifically, we show that our method of measuring the embedding space coverage has broad applicability across various encoders and datasets used for pretraining. We further observe that the fraction of buckets occupied by the representations saturates over time. These findings highlight that LSH is successful in capturing the differences between legitimate users and adversaries—even in a low-query regime. Finally, we note that our total number of buckets ($2^{12}$) is well calibrated since, over all datasets, it successfully maps multiple representations to the same hash bucket while still filling various fractions of the total number of buckets.

## 4.2 Calibrating the Cost Function

We experiment with different sets of hyperparameters to instantiate the cost function from Equation (1) in the previous section (3.3). As described there, we can calibrate the function (as shown in Figure 4) such that a desired penalty (in the form of a specific $\sigma$) will be assigned at a certain fraction of buckets occupied. For B4B, we aim at penalizing high embedding space coverage severely. Therefore, we need to identify and optimize for two components: 1) which value of $\sigma$ leads to significant performance drops, and 2) for what

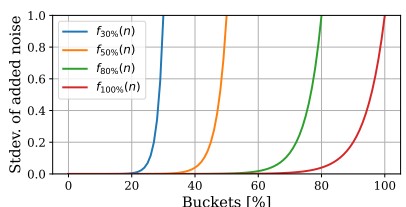

Figure 4: **Cost Function Calibration.**

fraction of coverage do we want to impose this significant drop. We base both components on empirical observations. Our first observation is that for our four downstream tasks (FashionMNIST, SVHN, STL10, and CIFAR10), performance drops to 10% (*i.e.,* random guessing) at roughly $\sigma = 0.5$. In Figure 3, we further see that with 50k queries, the downstream tasks occupy $< 30\%$ of the buckets. Ultimately, setting $\alpha$ and $\beta$ are design choices that an API provider needs to make in order to specify what type of query behavior they want to penalize. As very loose bounds (weak defense), based on our observation, we consider $\sigma = 1$ as a high penalty, which leads to $\alpha = 1$, and select $\beta = 0.8$. This $\beta$ corresponds to considering 80% of buckets filled as a too-large coverage of the embedding space. We empirically observe that coverage of 80% of buckets occurs, for example, after around 100k of ImageNet queries. By choosing our target $\beta$ so loose, *i.e.,* significantly larger than the observed 30% for downstream tasks, we offer flexibility for the API to also provide good representations for more complex downstream tasks. Finally, to obtain a flat cost curve close to the origin—which serves to map small fractions of covered buckets to small costs—we find that we can set $\lambda = 10^{-6}$. In the Appendix, we evaluate our defense end-to-end with differently parameterized cost functions.

### 4.3 Assessing the Effect of Transformations

**Transformations Do Not Harm Utility for Legitimate Users.** We evaluate the downstream accuracy for transformed representations based on training a linear classifier on top of them. To separate the effect of the noise added by our defense from the effect of the transformations, we perform the experiments in this subsection without adding noise to the returned representations. For example, on the CIFAR10 dataset and a SimSiam encoder pre-trained on ImageNet, without any transformations applied, we obtain a downstream accuracy of 90.41% ($\pm$ 0.02), while, with transformations, we obtain 90.24% ($\pm$ 0.11) for Affine, 90.40% ($\pm$ 0.05) for Pad+Shuffle, 90.18% ($\pm$ 0.06) for Affine+Pad+Shuffle, and 88.78% ($\pm$ 0.2) for Binary. This highlights that the transformations preserve utility for legitimate users. This holds over all datasets we evaluate as we show in Appendix F.

**Adversaries Cannot Perfectly Remap Representations over Multiple Sybil Accounts.** To understand the impact of our per-user account transformations on sybil-attack based encoder stealing, we evaluate the difficulty of remapping representations between different sybil accounts. For simplicity, and since we established in Section 3.4 that multi-account attacks reduce to a two-account setup, we assume an adversary who queries from two sybil accounts and aims at learning to map the transformed representations from account #2 to the representation space of account #1. Using more accounts for the adversary causes a larger query overhead and potentially more performance loss from remapping. Our evaluation here, hence, represents a lower bound on the overhead caused to the adversary through our transformations.

We learn the mapping between different accounts' representations by training a linear model on overlapping representations between the accounts. We assess the fidelity of remapped representations as a function of the number of overlapping queries between the accounts. As a fidelity metric for our remapping, we compare the cosine distance between representations ($a$ and $b$ defined as: $1 - \frac{a^T b}{||a||_2 \cdot ||b||_2}$). Once the remapping is trained, we evaluate by querying 10k data points from the test dataset through account #1 and then again through account #2. Then, we apply the learned remapping to the latter one and compute the pairwise cosine distances between the representations from account #1 and their remapped counterparts from account #2.

Our results are depicted in Figure 5. We show that the largest cosine distance is achieved with the binary transformations, making them the most protective against the adversary since they best prevent perfect remapping, even with an overlap of as many as 10k queries between both accounts. However, these binary transformations also incur the highest drop in accuracy for legitimate users. The defender has the possibility of selecting their preferred types of transformations between representations taking into account the trade-offs between the effectiveness of the defense and the negative impact on legitimate users.

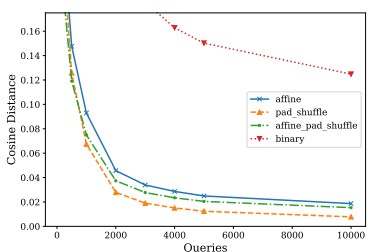

Figure 5: **Quality of Remappings.**

Table 1: **Stealing and Using Encoders With and Without our Defense.** The *USER* column represents the type of the APIs' user, where LEGIT denotes a legitimate user, ATTACKER stands for a standard single-account adversary, and SYBIL represents an adversary using two sybil accounts. We use InfoNCE loss for encoder extraction. # Queries stands for the number of queries used for stealing with ALL denoting that the entire downstream dataset was used. The *TYPE* column expresses how the dataset is used. We follow the stealing setup from [17]. In the first row, we present the undefended victim encoder's performance as the accuracy for downstream tasks trained on the encoder's returned representations. In the following row, we show downstream utility for legitimate users when the victim encoder is defended by our B4B. Finally, (in the remaining rows) we assess the performance of stolen encoders on the downstream tasks. Our results highlight that while the performance of the encoder for legitimate users stays high, our B4B renders stealing inefficient with the stolen encoders obtaining significantly worse performance on downstream tasks.

| USER | DEFENSE | # QUERIES | DATASET | TYPE | CIFAR10 | STL10 | SVHN | F-MNIST |
|---|---|---|---|---|---|---|---|---|
| LEGIT | NONE | ALL | TASK | QUERY | $90.41_{\pm 0.02}$ | $95.08_{\pm 0.13}$ | $75.47_{\pm 0.04}$ | $91.22_{\pm 0.11}$ |
| LEGIT | B4B | ALL | TASK | QUERY | $90.24_{\pm 0.11}$ | $95.05_{\pm 0.1}$ | $74.96_{\pm 0.13}$ | $91.7_{\pm 0.01}$ |
| ATTACK | NONE | 50K | IMGNET | STEAL | $65.2_{\pm 0.03}$ | $64.9_{\pm 0.01}$ | $63.1_{\pm 0.01}$ | $88.5_{\pm 0.01}$ |
| ATTACK | B4B | 50K | IMGNET | STEAL | $35.72_{\pm 0.04}$ | $31.54_{\pm 0.02}$ | $19.74_{\pm 0.02}$ | $70.01_{\pm 0.01}$ |
| ATTACK | NONE | 100K | IMGNET | STEAL | $68.1_{\pm 0.03}$ | $63.1_{\pm 0.01}$ | $61.5_{\pm 0.01}$ | $89.0_{\pm 0.07}$ |
| ATTACK | B4B | 100K | IMGNET | STEAL | $12.01_{\pm 0.07}$ | $13.94_{\pm 0.05}$ | $19.96_{\pm 0.03}$ | $69.63_{\pm 0.07}$ |
| ATTACK | NONE | 100K | LAION | STEAL | $64.92_{\pm 0.03}$ | $62.51_{\pm 0.03}$ | $59.02_{\pm 0.02}$ | $84.54_{\pm 0.01}$ |
| ATTACK | B4B | 100K | LAION | STEAL | $40.96_{\pm 0.03}$ | $40.69_{\pm 0.05}$ | $34.43_{\pm 0.01}$ | $72.92_{\pm 0.01}$ |
| SYBIL | B4B | 2×50K | IMGNET | STEAL | $39.56_{\pm 0.06}$ | $38.50_{\pm 0.04}$ | $23.41_{\pm 0.02}$ | $77.01_{\pm 0.08}$ |
| SYBIL | B4B | 3×33.3K | IMGNET | STEAL | $33.87_{\pm 0.05}$ | $38.57_{\pm 0.06}$ | $21.16_{\pm 0.01}$ | $72.95_{\pm 0.05}$ |
| SYBIL | B4B | 4×25K | IMGNET | STEAL | $33.98_{\pm 0.04}$ | $34.52_{\pm 0.08}$ | $21.21_{\pm 0.02}$ | $70.71_{\pm 0.05}$ |
| SYBIL | B4B | 5×20K | IMGNET | STEAL | $32.65_{\pm 0.05}$ | $32.45_{\pm 0.05}$ | $29.63_{\pm 0.01}$ | $70.12_{\pm 0.08}$ |
| SYBIL | B4B | 6×16.7K | IMGNET | STEAL | $26.62_{\pm 0.04}$ | $26.85_{\pm 0.05}$ | $24.32_{\pm 0.02}$ | $70.51_{\pm 0.04}$ |

## 4.4 End-to-End Stealing of an Encoder under our Defense

We perform an end-to-end study to showcase how our B4B defense affects legitimate users vs adversaries. The hyperparameters for B4B are chosen according to the empirical evaluation of the previous sections with $2^{12}$ as the number of buckets, $\alpha = 1, \beta = 0.8, \lambda = 10^{-6}$ as the hyperparameter of the cost function, and different random affine transformations per-user account. Our main results are presented in Table 1. We observe that instantiating our framework with B4B has a negligible impact on legitimate users while substantially lowering the performance of stolen encoders in the case of single-user and sybil attackers.

**Legitimate Users.** We compare the accuracy of downstream classifiers trained on top of unprotected vs defended encoders. The victim encoder achieves high accuracy on the downstream tasks when no defense is employed. With B4B in place, we observe that across all the downstream tasks, the drop in performance is below 1%. For example, there is only a slight decrease in the accuracy of CIFAR10 from 90.41±0.02% to 90.24±0.11%. B4B's small effect on legitimate users stems from the fact that their downstream representations cover a relatively small part of the representations space. This results in a very low amount of noise added to their representations which preserves performance.

**Adversaries.** For adversaries who create a stolen copy of the victim encoder, we make two main observations. The most crucial one is that when our B4B is in place, the performance of the stolen copies over all downstream tasks significantly drops in comparison to when the victim encoder is unprotected (grey rows in Table 1). This highlights that our B4B effectively prevents stealing. Our next key observation concerns the number of stealing queries used by the adversary: When no defense is applied, the more queries are issued against the API (*e.g.,* 100k instead of 50k), the higher performance of the stolen encoder on downstream tasks (*e.g.,* CIFAR10 or FashionMNIST). In contrast, with B4B implemented as a defense, the performance decreases when using more stealing queries from a single account. This is because with more queries issued, the coverage of embedding space grows which renders the returned representations increasingly noisy and harms stealing performance.

Moreover, we show that B4B can also prevent model stealing attacks with data from a different distribution than the victim encoder's training set. We highlight this in Table 1 where we also use the LAION-5B dataset to steal an ImageNet pre-trained encoder. Our results highlight first that without any defense in place, the LAION dataset is highly effective to extract the ImageNet pre-trained encoder. Second, B4B effectively defends against such attacks, and yields a significant drop in downstream accuracy (on average above 20%) of the stolen encoder.

We also show that this performance drop cannot be prevented by sybil attacks. Therefore, we first consider an adversary who queries from two sybil accounts with 50k queries issued per account and the first 10k queries of both accounts used to learn the remapping of representations between them. When the adversary trains their stolen encoder copy on all the remapped representations, they increase downstream performance over querying from a single account. Yet, their performance is still significantly smaller than the performance of the victim encoder for legitimate users, or the encoder stolen from an undefended victim. Moreover, using more than two sybil accounts further reduces the attack performance as remapping complications accumulate. With ten sybils, remapping leaves no more usable data for training the stolen encoder. This demonstrates our method's advantage: increasing the number of sybil accounts makes encoder extraction impractical due to the growing remapping overhead. Overall, the results highlight that our B4B also successfully prevents sybil attacks.

### 4.5 Baseline Comparison

Finally, we compare our B4B against the current state-of-the-art baseline defense, namely a static addition of noise to all the returned representations (as proposed in [16] (Section A.4),[29, 36]). For direct comparability, we evaluate the defense using the our end-to-end experiment setup from the previous section. We present our results in Table 6 in Appendix F.4. Confirming the findings from [16] our results also show that defenses that rely on static noise have the great disadvantage to harm legitimate users and attackers equally. When adding noise with a small standard deviation of $\sigma = 0.1$, we observe a negligible (<1%) performance drop for both attackers and legitimate users. Adding noise with a large standard deviation of, for example, $\sigma = 10$, we observe that both legitimate users' and attackers' performance drops between 15% and >40%. In summary, these defenses can either effectively defend stealing (but harm legitimate users), or keep utility for legitimate users high (but not defend well against stealing). In contrast, our B4B is able to provide high performance for legitimate users while effectively defending the encoder against stealing attacks.

## 5 Conclusions

We design B4B a new and modular active defense framework against stealing SSL encoders. All the previous approaches were either reactive, acting after the attack happened to detect stolen encoders, or lowered the quality of outputs substantially also for legitimate users which rendered such mechanisms impractical. We show that B4B successfully distinguishes between legitimate users and adversaries by tracking the embedding space coverage of users' obtained representations. B4B then leverages this tracking to apply a cost function that penalizes users based on the current space coverage, for instance, by lowering the quality of their outputs. Finally, B4B prevents sybil attacks by implementing per-user transformations for the returned representations. Through our experimental evaluation, we show that our defense indeed renders encoder stealing inefficient while preserving downstream utility for legitimate users. Our B4B is therefore a valuable contribution to a safer sharing and democratization of high-utility encoders over public APIs.

## Acknowledgments and Disclosure of Funding

This research was supported by Warsaw University of Technology within the Excellence Initiative Research University (IDUB) programme, National Science Centre, Poland grant no 2020/39/O/ST6/01478, grant no 2020/39/B/ST6/01511, grant no 2022/45/B/ST6/02817, and in part by PL-Grid Infrastructure grant nr PLG/2022/016058. The authors applied for a CC BY license to any Author Accepted Manuscript (AAM) version arising from this submission, in accordance with the grants' open access conditions.

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

# A  Broader Impacts

The goal of our work is to actively defend self-supervised encoders against model stealing attacks. Since we are directly defending encoders, any negative societal impacts of our work are minimal. One potentially negative impact could be the degradation of performance for legitimate users. However, as shown in our experimental results, we are able to preserve high utility for standard users.

# B  Limitations

We show how our defense method is tuned for SimSiam and DINO. There are more types of SSL encoders that can be tested with our method. The B4B defense method requires tuning the parameters, such as the number of occupied buckets that is allowed without any penalty for the cost function, or the selection of the transformations. These steps are rather difficult to automate but can be replaced with more data-driven approaches. For example, instead of designing a cost function from scratch, one could create an ML model to obtain a cost for a given occupation of the representation space. We explain more details in the Appendix C.2.

# C  Alternative Building Blocks to Instantiate B4B

While we present a reference implementation of B4B in our work that instantiates the three building blocks with (1) Local Sensitive Hashing, (2) Utility of the Representations, and (3) a set of concrete transformations, there exists a multitude of alternatives to concretely implement our B4B framework. In the following, we present these alternatives, grouped by building block.

## C.1  Alternative Estimation of the Coverage of Embedding Space

We also explore alternative methods to measure the distances between representations for queries sent to an API. One of them is to apply the cosine distance (where for two representations $a$ and $b$, it is defined as: $1 - \frac{a^T b}{||a||_2 \cdot ||b||_2}$) since it can be measured between individual data points in a pair-wise fashion. If the total pair-wise cosine distance between representations for a given user is small, then the user queries presumably come from a single downstream task distribution. Otherwise, a user might be malicious and would like to cover a large part of the representation space, then the total pair-wise cosine distance for the user's representations would be high. Note that in this case, the cosine distance can be replaced with any other distance measure, such as the Minkowski distance. We opt for the LSH in our reference implementation, since it is much less expensive to compute than cosine distance. LSH requires only $2^{12} = 4096$ buckets that can be expressed as a binary table with the same number of elements, which requires in the worst case iterating over all of them to count how many are occupied. With more than $4096$ queries sent by a given user, the computation on the LSH is sublinear $< O(n)$ with respect to the number of user queries. For the cosine distance approach, the required computation grows quadratically $O(n^2)$ with the number of queries.

## C.2  Alternative Cost Functions

The cost functions can be designed from scratch manually or learned, for example, via an ML model, such as a neural network or SVM. In our initial version, the function was designed manually, where the underlying premise is that once a specified number of buckets is occupied, the cost should grow exponentially. Instead of defining such a function or providing the high-level parameters for functions that we contributed, one could learn an ML model that for a given number of buckets occupied, it should output an estimated cost, or even directly, the desired $\sigma$ (standard deviation) of the noise added to representations. This method requires a relatively large number of data points to be provided for training the model, however, lowers the burden on a defender to either decide on the specific function or adjust its parameters. Thus, it could be more user-friendly, for example, not necessitating any mathematical background, but can be precise enough to obtain the desired behavior.

Note that instead of adding the calibrated noise (proportional to the estimated cost) to the representations, we could rather require a given user to pay a higher monetary cost for queries that cover a large fraction of the representation space, or force a user to solve a puzzle in a form of the proof-of-work [18], wait a specified amount of time via proof-of-elapsed time (PoET) [4], or prove that

a specified amount of disk space was reserved [19, 20]. For example, consider the approach with PoET. A user sends queries to the API, which we cost based on their occupation of the embedding space. The user is sent a waiting time. The users' resource (e.g., a CPU) has to be occupied for this specific waiting time without performing any work. At the end of the specified amount of time, the user sends proof of the elapsed time, which can be easily verified by the server. PoET requires access to specialized hardware, for example, secure CPU instructions that are becoming widely available in consumer and enterprise processors. If a user does not own such hardware, proof of elapsed time could be produced using a service exposed by a cloud provider (e.g., Azure VMs that feature TEE 2). Note that if a server sends the time based on the calculated cost, the adversary might learn the cost function. Instead, the exact waiting time should be split in random *subwaiting* times and sent to the user one by one. Thus, a server should rather have a few rounds of exchange with the client to incur the additional cost.

## C.3    Alternative to Transformations

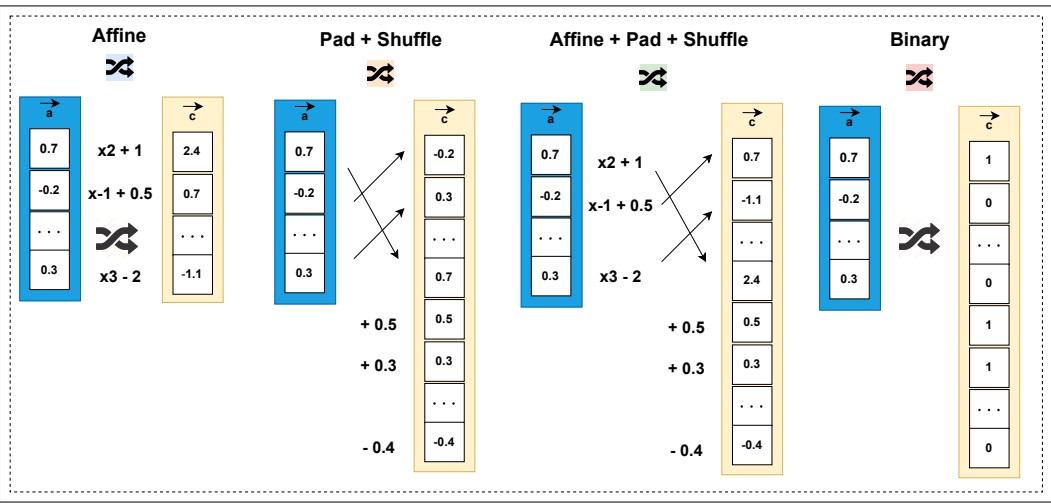

Figure 6: **Overview on Transformations.** We depict the inner-workings of the transformations considered in this work.

As an alternative to the transformations used within this work (see Figure 6), one could use a different set of transformations or combinations thereof. The padding can be done with different constant values and combined with adding constant values within the representations. The padding and adding the constant values can be followed by shuffling the elements within the representations. We can apply the affine of binary transformations on top of the padding and shuffling. Additionally, we can also use other pre-defined linear transformations like rotations or shearing.

The representations could also be compressed to smaller vectors and the compression rate would depend on the occupation of the representation space, for example, the higher the number of occupied buckets in our hash table, the more compressed the output representations could be. Such representations could be compressed via FFT, a cosine transform, or standard compression techniques such as snappy [5]. If the information from the representations should not be lost, then the lossless compression techniques can be applied, for instance, zstd [6]. The only requirement of the compression techniques is to ensure that they do not decrease the accuracy on downstream tasks for legitimate users.

Another alternative is to incorporate an additional neural network layer for transforming the returned representations. The training of this supplementary layer should primarily focus on preserving the usability of the representations for legitimate users. This approach grants the API provider with additional capabilities, as it allows for the utilization of customized training objectives. For instance, if the API provider employs LSH (Locality-Sensitive Hashing) to estimate the coverage of the representation space, they can leverage buckets and train the additional layer to maintain high-quality representations exclusively for frequently-used buckets and their surrounding areas, while not prioritizing the rest of the representation space. This approach safeguards legitimate users

from any adverse effects, as their coverage of the representation space is minimal. Simultaneously, it ensures that adversaries are unable to exploit representations from the entire representation space.

## D  Sybil Attacks

We consider an adversary who generates $n$ sybil accounts to steal the encoder from the API. For each of the accounts, the representations are transformed in a different way. Therefore, to replicate the victim model using all the obtained representations, the adversary has to map these representations into one single space. This can be done, for example, by training a neural network to perform the mapping.

We assume the adversary obtains $\{N_1, N_2, \ldots, N_n\}$ many representations from the victim for each of the $n$ sybil accounts. Without loss of generality, we assume the adversary maps them back to the embedding space of the first sybil account. To learn the mapping, the adversary can apply different strategies.

### D.1  Sybil Strategies

We present three potential approaches that Sybils might want to apply to circumvent our defense. Consider three users: $A$, $B$, and $C$, with their respective datasets $D_A$, $D_B$, and $D_C$, each with different distributions to maximize extraction effectiveness. First, user $A$ is selected to unify representations from other users $B$ and $C$. User $A$ would have to query from at least two different datasets $D_B$ and $D_C$, while other users would act legitimately. Sybil attackers want to deploy as many users as possible but with more fake accounts, user $A$ incurs high coverage of the representation space, and this is prevented by our single-user defense. In all other cases neither of the sybil users can act legitimately, thus they are already affected by the single-user defense. Second, user $A$ would query from their own dataset $D_A$ and partially from dataset $D_B$. Then user $B$ would query from their own dataset $D_B$ and partially from dataset $D_C$, and so on. This method is the most inconspicuous but requires a number of remappings that scales super-exponentially with the number of fake accounts, which is impractical. Finally, each user would query from their respective dataset, for example, user $B$ would query from dataset $D_B$ and additionally from a remapping dataset, *e.g.,* $D_A$. Representations could be unified by mapping them to $A$'s representations. The last approach as well as all other cases reduce to the minimum of remapping between representations of a pair of users. We show that our defense cuts such attempts short by ensuring that the remapping between representations is prohibitive even for a pair of users.

## E  Additional Related Work

One of the main workhorse techniques used in the encoders is contrastive learning, where the representations are trained so that the positive pairs (two augmented versions of the same image) have similar representations while negative pairs (augmentations of two different images) have representations which are far apart.

**SimSiam** utilizes Siamese networks (two encoders with shared weights) but with a simplified training process and architecture. In contrast to the previous frameworks, such as SimCLR [10], SimSiam's authors show that negative samples are unnecessary and collapsing solutions can be avoided by applying the projection head to of one of the encoders, and a stop-gradient operation to the other. SimSiam minimizes the negative cosine similarity between two randomly augmented views of the same image from the Siamese encoders, which is expressed via a symmetrized loss [22]. This creates a simple yet highly effective representation learning method.

**DINO** is another popular representation learning framework. While SimSiam uses CNNs, DINO employs vision transformers (ViTs). It trains a student and teacher encoder with the same architecture, updating the teacher with an (exponential moving) average of the student. Different random transformations of the same image are passed through both encoders. The student receives smaller image crops, forcing it to generate representations restoring parts of the original image. The training objective is minimizing cross-entropy loss between teacher and student representations.

# F Additional Experimental Results

## F.1 Details on Experimental Setup

The end-to-end experiments on stealing SimSiam and ViT DINO encoders were done using 3 A100 GPUs. Detailed experiments including mapping, transformations and the evaluation was performed using a single computer equipped with two Nvidia RTX 2080 Ti GPUs.

## F.2 Datasets Used

**CIFAR10** [28]: The CIFAR10 dataset consists of 32x32 colored images with 10 classes. There are 50000 training images and 10000 test images.

**SVHN** [32]: The SVHN dataset contains 32x32 coloured images with 10 classes. There are roughly 73000 training images, 26000 test images and 530000 "extra" images.

**ImageNet**[15]: Larger sized coloured images with 1000 classes. As is commonly done, we resize all images to be of size 224x224. There are approximately 1 million training images and 50000 test images.

**STL10** [13]: The STL10 dataset contains 96x96 coloured images with 10 classes. There are 5000 training images, 8000 test images, and 100000 unlabeled images.

**LAION-5B** [35] The LAION-5B dataset consists of 5,85 billion CLIP-filtered image-text pairs. The dataset was crawled from publically available internet.

## F.3 More Results for the End2End Empirical Evaluation

We consider fine-tuning parameters $\beta$, $\lambda$, and $\alpha$ for our cost function and the intuitive meaning behind these parameters. In general, our recommendation is to adjust the parameter $\beta$ that specifies how many buckets are allowed to be filled by users' downstream tasks. On the other hand, when parameter $\lambda$ is increased, this causes a higher amount of added noise before we reach the number of buckets specified by $\beta$, which lowers the performance of a given downstream task relatively early. For example, a higher value of $\lambda$ in Figure 4, would cause an increase in the amount of added noise much earlier than for the target value of $\beta$. Finally, parameter $\alpha$ controls the amount of noise once the number of buckets specified by $\beta$ is reached. Thus, in Figure 4, we set $\alpha = 1$ and the maximum standard deviation of the added Gaussian noise is 1.

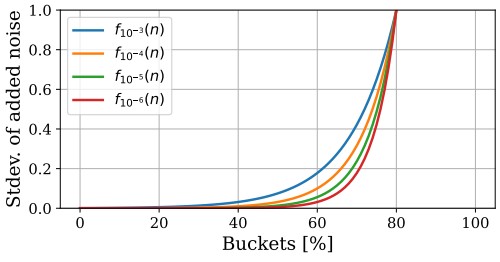
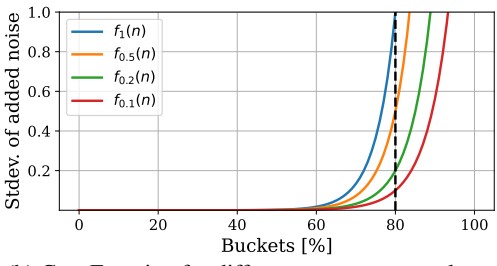

(a) Cost Function for different $\lambda$ parameter values.  (b) Cost Function for different $\alpha$ parameter values.

Figure 7: **Effects of $\lambda$ and $\alpha$ parameters on the Cost Function.** We present the Cost Function for $\alpha$=1, $\beta$=80 and different values of $\lambda$ (*left*) and $\lambda = 10^{-6}$, $\beta$=80 and different values of $\alpha$ (*right*).

Table 2: **Stealing and Using Encoders With and Without our Defense**. The model used in the experiments is Simsiam, with the following parameters for the cost function $\lambda = 10^{-4}$, $\alpha = 1$, and $\beta = 80\%$, and the number of buckets equal to $2^{12}$. Due to the higher value of the parameter $\lambda$, we observe lower performance on downstream tasks for the attackers since the magnitude of noise added to the representations is higher. However, for more complicated tasks than CIFAR10, this change might cause a potential drop in accuracy for the legitimate users.

| USER | DEFENSE | # QUERIES | DATASET | TYPE | CIFAR10 | STL10 | SVHN | F-MNIST |
|---|---|---|---|---|---|---|---|---|
| LEGIT | NONE | ALL | TASK | QUERY | $90.41_{\pm 0.02}$ | $95.08_{\pm 0.13}$ | $75.47_{\pm 0.04}$ | $91.22_{\pm 0.11}$ |
| LEGIT | B4B | ALL | TASK | QUERY | $90.02_{\pm 0.1}$ | $94.88_{\pm 0.17}$ | $74.72_{\pm 0.13}$ | $91.76_{\pm 0.09}$ |
| ATTACK | NONE | 50K | IMGNET | STEAL | $65.2_{\pm 0.03}$ | $64.9_{\pm 0.01}$ | $62.1_{\pm 0.01}$ | $88.5_{\pm 0.01}$ |
| ATTACK | B4B | 50K | IMGNET | STEAL | $28.22_{\pm 0.04}$ | $26.62_{\pm 0.02}$ | $19.62_{\pm 0.02}$ | $78.41_{\pm 0.01}$ |
| ATTACK | NONE | 100K | IMGNET | STEAL | $68.1_{\pm 0.03}$ | $63.1_{\pm 0.01}$ | $61.5_{\pm 0.01}$ | $89.0_{\pm 0.07}$ |
| ATTACK | B4B | 100K | IMGNET | STEAL | $17.73_{\pm 0.18}$ | $15.59_{\pm 0.61}$ | $19.53_{\pm 0.01}$ | $55.11_{\pm 0.05}$ |
| SYBIL | B4B | 50K+50K | IMGNET | STEAL | $33.43_{\pm 0.03}$ | $31.18_{\pm 0.12}$ | $22.91_{\pm 0.01}$ | $75.35_{\pm 0.05}$ |

Table 3: **Stealing and Using Encoders With and Without our Defense**. The model used in the experiments is Simsiam, with the following parameters for the cost function $\lambda = 10^{-6}$, $\alpha = 1$, and $\beta = 50\%$, and the number of buckets equal to $2^{12}$. This experiment corresponds to considering 50% of buckets filled as a too-large coverage of the embedding space. This improves the defense but again might potentially harm the performance of more complicated tasks than CIFAR10 since they could occupy more buckets than 50%.

| USER | DEFENSE | # QUERIES | DATASET | TYPE | CIFAR10 | STL10 | SVHN | F-MNIST |
|---|---|---|---|---|---|---|---|---|
| LEGIT | NONE | ALL | TASK | QUERY | $90.41_{\pm 0.02}$ | $95.08_{\pm 0.13}$ | $75.47_{\pm 0.04}$ | $91.22_{\pm 0.11}$ |
| LEGIT | B4B | ALL | TASK | QUERY | $90.27_{\pm 0.07}$ | $95.12_{\pm 0.13}$ | $74.94_{\pm 0.16}$ | $91.66_{\pm 0.05}$ |
| ATTACK | NONE | 50K | IMGNET | STEAL | $65.2_{\pm 0.03}$ | $64.9_{\pm 0.01}$ | $62.1_{\pm 0.01}$ | $88.5_{\pm 0.01}$ |
| ATTACK | B4B | 50K | IMGNET | STEAL | $15.52_{\pm 0.37}$ | $12.57_{\pm 0.23}$ | $19.53_{\pm 0.01}$ | $23.17_{\pm 0.01}$ |
| ATTACK | NONE | 100K | IMGNET | STEAL | $68.1_{\pm 0.03}$ | $63.1_{\pm 0.01}$ | $61.5_{\pm 0.01}$ | $89.0_{\pm 0.07}$ |
| ATTACK | B4B | 100K | IMGNET | STEAL | $16.27_{\pm 0.04}$ | $13.93_{\pm 0.35}$ | $19.54_{\pm 0.02}$ | $54.69_{\pm 0.02}$ |
| SYBIL | B4B | 50K+50K | IMGNET | STEAL | $30.14_{\pm 0.01}$ | $29.57_{\pm 0.08}$ | $19.99_{\pm 0.03}$ | $71.72_{\pm 0.01}$ |

Table 4: **Stealing and Using Encoders With and Without our Defense**. The model used in the experiments is Simsiam, with the following parameters for the cost function $\lambda = 10^{-6}$, $\alpha = 1$, and $\beta = 30\%$, and the number of buckets equal to $2^{12}$. Since the value of parameter $\beta$ is decreased substantially to 30%, we observe a drop in accuracy for legitimate users. For example, more than 1% for CIFAR10. In the next Table 5, we show that by also decreasing the parameter $\alpha$, we can attenuate this harmful effect and retain higher accuracy for legitimate users. In case of an attack, for 100k stealing queries, we observe much lower accuracy levels than for $\beta = 50\%$ shown in Table 3.

| USER | DEFENSE | # QUERIES | DATASET | TYPE | CIFAR10 | STL10 | SVHN | F-MNIST |
|---|---|---|---|---|---|---|---|---|
| LEGIT | NONE | ALL | TASK | QUERY | $90.41_{\pm 0.02}$ | $95.08_{\pm 0.13}$ | $75.47_{\pm 0.04}$ | $91.22_{\pm 0.11}$ |
| LEGIT | B4B | ALL | TASK | QUERY | $88.1_{\pm 0.11}$ | $94.92_{\pm 0.11}$ | $74.37_{\pm 0.02}$ | $91.67_{\pm 0.07}$ |
| ATTACK | NONE | 50K | IMGNET | STEAL | $65.2_{\pm 0.03}$ | $64.9_{\pm 0.01}$ | $62.1_{\pm 0.01}$ | $88.5_{\pm 0.01}$ |
| ATTACK | B4B | 50K | IMGNET | STEAL | $30.82_{\pm 0.09}$ | $26.37_{\pm 0.07}$ | $21.87_{\pm 0.03}$ | $66.0_{\pm 0.02}$ |
| ATTACK | NONE | 100K | IMGNET | STEAL | $68.1_{\pm 0.03}$ | $63.1_{\pm 0.01}$ | $61.5_{\pm 0.01}$ | $89.0_{\pm 0.07}$ |
| ATTACK | B4B | 100K | IMGNET | STEAL | $9.57_{\pm 0.17}$ | $9.83_{\pm 0.09}$ | $19.57_{\pm 0.01}$ | $27.06_{\pm 0.46}$ |
| SYBIL | B4B | 50K+50K | IMGNET | STEAL | $29.15_{\pm 0.02}$ | $28.67_{\pm 0.06}$ | $19.98_{\pm 0.03}$ | $70.62_{\pm 0.03}$ |

Table 5: **Stealing and Using Encoders With and Without our Defense**. The model used in the experiments is Simsiam, with the following parameters for the cost function $\lambda = 10^{-6}$, $\alpha = 0.1$, and $\beta = 30\%$, and the number of buckets equal to $2^{12}$. Due to the lower performance on downstream tasks observed in Table 4 while keeping the parameter $\beta$ fixed to 30% and $\lambda$ fixed to $10^{-6}$, we decrease the value of parameter $\alpha$ to 0.1, which increases the performance of legitimate users on their downstream tasks. In this experiment, we also carry out a sybil attack with more accounts (4 instead of 2), but observe that this modification does not improve the performance of the attacker. With more accounts, a sybil has to sacrifice more queries for the remappings between the representations from different accounts. Additionally, note that each account introduces a different remapping error by the dint of different transformations applied to each account by B4B.

| USER | DEFENSE | # QUERIES | DATASET | TYPE | CIFAR10 | STL10 | SVHN | F-MNIST |
|------|---------|-----------|---------|------|---------|-------|------|---------|
| LEGIT | NONE | ALL | TASK | QUERY | $90.41_{\pm0.02}$ | $95.08_{\pm0.13}$ | $75.47_{\pm0.04}$ | $91.22_{\pm0.11}$ |
| LEGIT | B4B | 50K | CIFAR10 | QUERY | $90.17_{\pm0.1}$ | $94.92_{\pm0.09}$ | $74.97_{\pm0.13}$ | $91.71_{\pm0.08}$ |
| ATTACK | NONE | 50K | IMGNET | STEAL | $65.2_{\pm0.03}$ | $64.9_{\pm0.01}$ | $62.1_{\pm0.01}$ | $88.5_{\pm0.01}$ |
| ATTACK | B4B | 50K | IMGNET | STEAL | $19.95_{\pm0.19}$ | $15.54_{\pm0.34}$ | $19.57_{\pm0.01}$ | $23.50_{\pm0.19}$ |
| ATTACK | NONE | 100K | IMGNET | STEAL | $68.1_{\pm0.03}$ | $63.1_{\pm0.01}$ | $61.5_{\pm0.01}$ | $89.0_{\pm0.07}$ |
| ATTACK | B4B | 100K | IMGNET | STEAL | $10.35_{\pm0.19}$ | $12.37_{\pm0.69}$ | $19.34_{\pm0.01}$ | $68.93_{\pm0.17}$ |
| SYBIL | B4B | 4×25K | IMGNET | STEAL | $33.15_{\pm0.04}$ | $30.23_{\pm0.07}$ | $20.87_{\pm0.01}$ | $72.19_{\pm0.02}$ |

## F.4 B4B vs Static Noise Addition Defenses

We compare our B4B against the current state-of-the-art baseline defense, namely adding a static addition of noise to all the returned representations (as proposed in [16] (Section A.4),[29, 36]). For the Table 6, we use the same setup as in Table 1 (with an ImageNet pre-trained encoder).

Our results show the following insights:

1. If the amount of noise is small ($\sigma = 0.1$) then the performance drop is negligible but for both a legitimate user (row 2) and an adversary (row 6). In this case, the defense does not affect the adversary at all (compare rows 5 & 6).

2. If the amount of noise is large ($\sigma = 10$) then the performance drop is large for both a legitimate user (row 3) and an adversary (row 7). In this case, the encoder is worthless for legitimate users since the performance is too low.

Table 6: **Stealing and Using Encoders with Static Noise Addition Defenses vs. Our B4B Defense.** Adding a small amount of noise results in negligible drop in performance for both legitimate user (row 2) and an adversary (row 6). Adding a large amount of noise defend stealing (row 7), but significantly harm legitimate users at the same time (row 3). Our B4B defense solves the above problem and provides high performance for legitimate users (row 4) while effectively defending the encoder against stealing attacks (row 8).

| USER | DEFENSE | # QUERIES | DATASET | TYPE | CIFAR10 | STL10 | SVHN | F-MNIST |
|------|---------|-----------|---------|------|---------|-------|------|---------|
| LEGIT | NONE | ALL | TASK | QUERY | $90.41_{\pm0.02}$ | $95.08_{\pm0.13}$ | $75.47_{\pm0.04}$ | $91.22_{\pm0.11}$ |
| LEGIT | NOISE $\sigma=0.1$ | ALL | TASK | QUERY | $90.20_{\pm0.03}$ | $95.15_{\pm0.13}$ | $75.29_{\pm0.09}$ | $91.24_{\pm0.02}$ |
| LEGIT | NOISE $\sigma=10$ | ALL | TASK | QUERY | $65.11_{\pm0.45}$ | $76.37_{\pm0.14}$ | $33.23_{\pm0.09}$ | $65.83_{\pm0.13}$ |
| LEGIT | B4B | ALL | TASK | QUERY | $90.24_{\pm0.11}$ | $95.05_{\pm0.1}$ | $74.96_{\pm0.13}$ | $91.7_{\pm0.01}$ |
| ATTACK | NONE | 50K | IMGNET | STEAL | $65.2_{\pm0.03}$ | $64.9_{\pm0.01}$ | $63.1_{\pm0.01}$ | $88.5_{\pm0.01}$ |
| ATTACK | NOISE $\sigma=0.1$ | 50K | IMGNET | STEAL | $64.92_{\pm0.04}$ | $64.61_{\pm0.02}$ | $62.35_{\pm0.01}$ | $88.41_{\pm0.01}$ |
| ATTACK | NOISE $\sigma=10$ | 50K | IMGNET | STEAL | $36.32_{\pm0.2}$ | $32.59_{\pm0.06}$ | $20.59_{\pm0.01}$ | $74.94_{\pm0.02}$ |
| ATTACK | B4B | 50K | IMGNET | STEAL | $35.72_{\pm0.04}$ | $31.54_{\pm0.02}$ | $19.74_{\pm0.02}$ | $70.01_{\pm0.01}$ |

### F.5 Additional Embedding Space Coverage experiments

We present additional experiments on measuring the coverage of the representation space.

First, we use the same set-up as from Table 1 - SimSiam with ResNet50 pretrained on ImageNet. When querying the encoder with ImageNet-Full (includes all 1000 classes) and LAION-5B datasets, they both occupy a large fraction of the representation space of the victim encoder, as shown on Figure 8. In contrast, CIFAR10 covers the smallest portion of the representation space as the simplest dataset tested. ImageNet-Dogs (with only 118 classes for dog breeds) falls in the middle, occupying more space than CIFAR10 but less than ImageNet-Full and LAION-5B. Its intermediate coverage aligns with its mid-level difficulty compared to the other datasets. As indicated by representation space coverage, stealing the encoder is similarly effective with ImageNet-Full and LAION-5B datasets, as both datasets cover a large fraction of the representation space. Overall, Figure 8 demonstrates that: 1) our B4B can successfully protect the encoder model even from attackers stealing with data that was not used to train the model (LAION-5B in this case) and 2) while providing clean representation for users querying from downstream tasks that are part of more complicated datasets (ImageNet-Dogs).

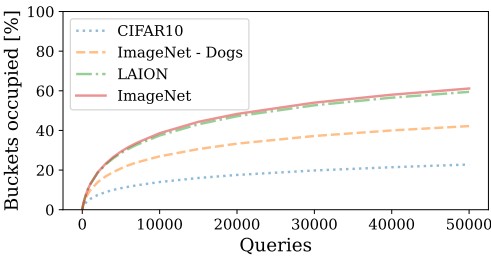

Figure 8: **Fraction of Occupied Buckets (Embedding Space Coverage) for the ImageNet encoder.** Representations for the downstream datasets (CIFAR10, ImageNet - Dogs) occupy a smaller fraction of buckets than representations from the complex ImageNet or LAION-5B datasets. The underlying encoder is SimSiam pre-trained on ImageNet with ResNet50.

Our method of measuring the embedding space coverage is not limited to a particular encoder or dataset used for pretraining. We demonstrate this in Figure 9, showing the fraction of occupied buckets for a SimCLR [11] Resnet34 encoder pretrained on CIFAR10.

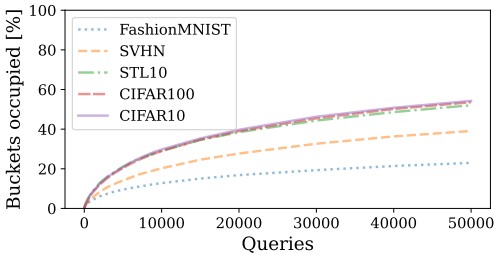

Figure 9: **Fraction of Occupied Buckets (Embedding Space Coverage) for the CIFAR10 encoder.** B4B can be applied to an encoder trained on CIFAR10. Representations for the downstream datasets (FashionMNIST, SVHN) occupy a smaller fraction of buckets than representations from CIFAR10, CIFAR100, and STL10 datasets. The underlying encoder is SimCLR pre-trained on CIFAR10 with ResNet34.

## F.6 Setting the number of buckets

We present our procedure to find an optimal number of buckets in Figure 10.

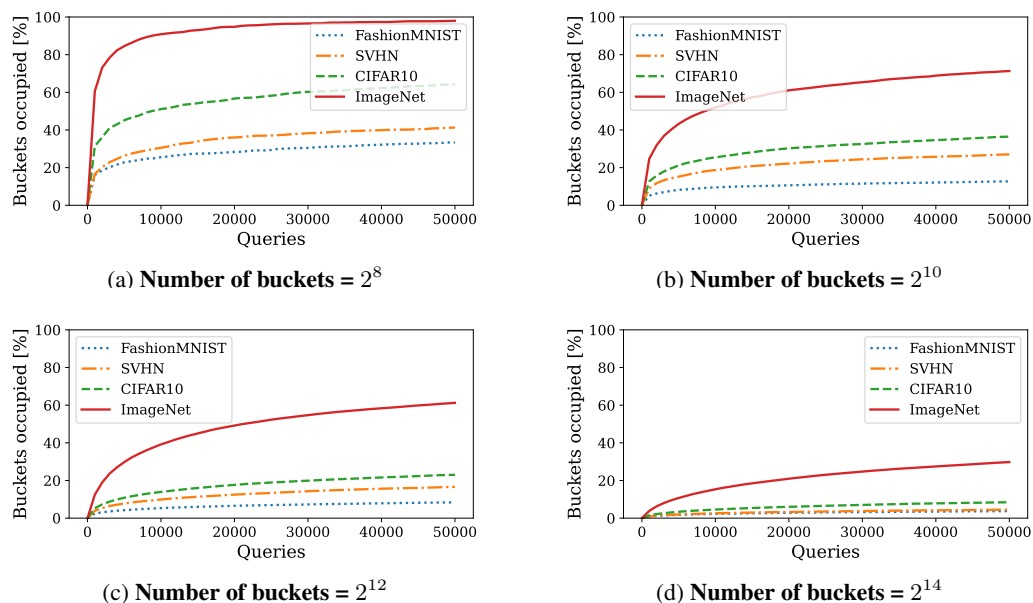

(a) **Number of buckets = $2^8$**

(b) **Number of buckets = $2^{10}$**

(c) **Number of buckets = $2^{12}$**

(d) **Number of buckets = $2^{14}$**

Figure 10: **Estimating Embedding Space Coverage through LSH on the SimSiam Encoder.** We extend the results from Figure 3(a) and present the fraction of buckets occupied by representations of different datasets as a function of the number of queries posed to the encoder. We consider different number of buckets in the LSH table. We observe that $2^8$ buckets is to small since queries from the ImageNet dataset saturate all the buckets after around 50k queries, while the number $2^{14}$ of buckets is too large since it is never occupied more than 40%. Thus, the number $2^{12}$ buckets is a good middle ground. Subfigure (c) corresponds to Figure 3 from the main paper. We also use the same notation and carry out our experiments in the same way as in Figure 3.

## F.7 Results for DINO

We show that our defense is also applicable to the DINO encoder. The occupation of the representations space is presented visually in Figure 11. We also show that the number of buckets $2^{12}$ is optimal for DINO in Figure 12. The impact of transformation on the representations from DINO is shown Table 8. Finally, the end to end experiment for DINO is presented in Table 7.

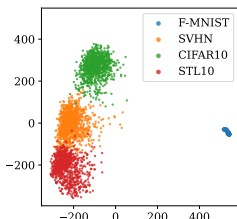

Figure 11: **Representations from Different Tasks Occupy Different Sub-Spaces of the Embedding Space. Presented for FashionMNIST, SVHN, CIFAR10, and STL10.** In this plot, we used the DINO ViT Small encoder trained on ImageNet.

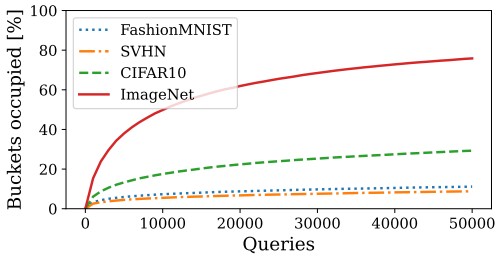

Figure 12: **Estimating Embedding Space Coverage through LSH on the DINO Encoder.** The number of buckets is set to $2^{12}$. We also use the same notation and carry out our experiments in the same way as in Figure 3.

Table 7: **Stealing and Using Encoders With and Without our Defense**. The model used in the experiments is DINO, with the following parameters for the cost function $\lambda = 10^{-6}$, $\alpha = 1000$, and $\beta = 60\%$, and the number of buckets equal to $2^{12}$. We have to increase the value of parameter $\alpha$ by $\times 1000$ since the norms of the DINO representations are also around $10^3$ higher than for SimSiam. We observe that B4B performs similarly on DINO as for SimSiam.

| USER | DEFENSE | # QUERIES | DATASET | TYPE | CIFAR10 | STL10 | SVHN | F-MNIST |
|---|---|---|---|---|---|---|---|---|
| LEGIT | NONE | ALL | TASK | QUERY | $94.51_{\pm 0.08}$ | $97.98_{\pm 0.04}$ | $70.66_{\pm 0.16}$ | $89.98_{\pm 0.03}$ |
| LEGIT | B4B | ALL | TASK | QUERY | $94.25_{\pm 0.11}$ | $98.05_{\pm 0.04}$ | $69.66_{\pm 0.14}$ | $89.68_{\pm 0.01}$ |
| ATTACK | NONE | 50K | IMGNET | STEAL | $67.92_{\pm 0.04}$ | $66.02_{\pm 0.22}$ | $61.30_{\pm 0.01}$ | $89.46_{\pm 0.01}$ |
| ATTACK | B4B | 50K | IMGNET | STEAL | $42.02_{\pm 0.05}$ | $38.91_{\pm 0.06}$ | $19.94_{\pm 0.02}$ | $73.33_{\pm 0.04}$ |
| ATTACK | NONE | 100K | IMGNET | STEAL | $75.07_{\pm 0.01}$ | $76.32_{\pm 0.02}$ | $71.79_{\pm 0.06}$ | $89.76_{\pm 0.01}$ |
| ATTACK | B4B | 100K | IMGNET | STEAL | $19.27_{\pm 0.03}$ | $21.24_{\pm 0.03}$ | $19.84_{\pm 0.01}$ | $71.01_{\pm 0.03}$ |
| SYBIL | B4B | 50K+50K | IMGNET | STEAL | $45.56_{\pm 0.06}$ | $42.50_{\pm 0.02}$ | $24.25_{\pm 0.03}$ | $78.01_{\pm 0.08}$ |

### F.8 Additional evaluation of transformations

Additionally, we show the impact of transformations on the performance of legitimate users in Table 8 (for both SimSiam and DINO).

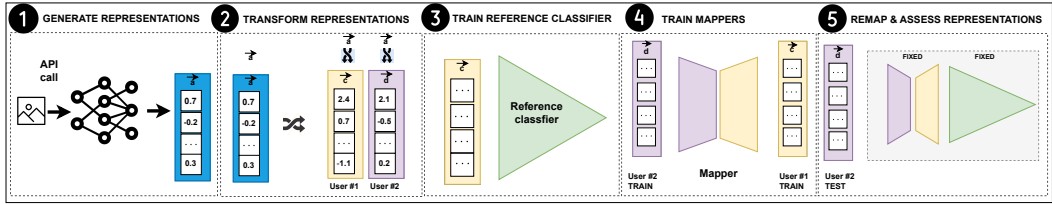

Figure 13: **Protocol to Evaluate the Mapping Between Representations.** We present the protocol of evaluating remappings for two sybil accounts. ❶ API receives inputs from two sybil accounts and generates corresponding representations. ❷ Representations are transformed on a per-user basis and returned. ❸ Adversary trains a reference classifier on representations from account one. ❹ Adversary trains a linear model to find mapping from representations of account two to representations of account one. ❺ To check the quality of obtained mapping representations from test set of account two are mapped using the fixed mapper (from step 4) to representation space of account one. This enables the calculation of cosine distance between representations from account one and their counterparts from account two shown in Figure 5. Additionally, the fixed reference classifier (from step 3) can be used to measure the accuracy drop caused by remapping .

Table 8: **Impact of Transformations on the Performance for Legitimate Users.** We show that the transformations applied per-account do not harm the performance of legitimate users on their downstream tasks. The victim encoders was trained on the ImageNet dataset using SimSiam and DINO frameworks.

| TRANSFORMATION | ENCODER | CIFAR10 | STL10 | SVHN | F-MNIST |
|---|---|---|---|---|---|
| NONE | *Victim SimSiam* | $90.41_{\pm 0.02}$ | $95.08_{\pm 0.13}$ | $75.47_{\pm 0.04}$ | $91.22_{\pm 0.11}$ |
| AFFINE | SIMSIAM | $90.24_{\pm 0.11}$ | $95.05_{\pm 0.1}$ | $74.96_{\pm 0.18}$ | $91.42_{\pm 0.15}$ |
| PAD+SHUFFLE | SIMSIAM | $90.4_{\pm 0.05}$ | $95.34_{\pm 0.06}$ | $75.47_{\pm 0.01}$ | $91.38_{\pm 0.15}$ |
| AFFINE+PAD+SHUFFLE | SIMSIAM | $90.18_{\pm 0.06}$ | $95.03_{\pm 0.05}$ | $74.86_{\pm 0.1}$ | $91.35_{\pm 0.1}$ |
| BINARY | SIMSIAM | $88.78_{\pm 0.2}$ | $94.72_{\pm 0.02}$ | $68.42_{\pm 0.16}$ | $88.91_{\pm 0.34}$ |
| NONE | *Victim DINO* | $94.51_{\pm 0.08}$ | $97.98_{\pm 0.04}$ | $70.66_{\pm 0.16}$ | $89.98_{\pm 0.03}$ |
| AFFINE | DINO | $94.25_{\pm 0.11}$ | $98.05_{\pm 0.04}$ | $69.77_{\pm 0.11}$ | $89.68_{\pm 0.01}$ |
| PAD+SHUFFLE | DINO | $94.72_{\pm 0.02}$ | $98.07_{\pm 0.03}$ | $70.44_{\pm 0.1}$ | $89.91_{\pm 0.08}$ |
| AFFINE+PAD+SHUFFLE | DINO | $94.26_{\pm 0.06}$ | $98.02_{\pm 0.01}$ | $69.49_{\pm 0.2}$ | $89.70_{\pm 0.1}$ |
| BINARY | DINO | $92.96_{\pm 0.1}$ | $98.03_{\pm 0.03}$ | $59.53_{\pm 0.27}$ | $88.26_{\pm 0.04}$ |

