# OpenReview forum: "Bucks for Buckets (B4B): Active Defenses Against Stealing Encoders"
_NeurIPS.cc/2023/Conference — NeurIPS 2023 poster_

### Official Review · Reviewer_pRsJ · 2023-06-19

**Soundness:** 2 fair
**Presentation:** 2 fair
**Contribution:** 2 fair
**Rating:** 5
**Confidence:** 4

**Summary:**

This paper studies active defenses against model stealing attacks, with a specific focus on SSL encoders. The authors propose B4B, a framework including three modules: 1) a coverage estimation to distinguish legitimate users and adversaries; 2) a cost function that maps the coverage to a penalty to prevent stealing; 3) a per-user transformation function to prevent sybil attacks. The results show that B4B can prevent stealing and also preserve the utility of representations for legitimate users.

**Strengths:**

- Trendy topic
- Extensive ablation studies

**Weaknesses:**

- Absence of comparison with previous work.
- Impractical assumption
- Design intuition lacks persuasiveness.
- Lack of experiments

**Questions:**

see limitations

**Limitations:**

This paper focuses on a trendy topic: defenses against model stealing attacks. To tackle this problem, the authors propose an active defense framework, namely B4B. They evaluate its performance on two SSL vision encoders, i.e., SimSiam and DINO, and consider four downstream datasets, i.e., CIFAR10, STL10, SVHN, and FashionMNIST.

I appreciate the authors' efforts in conducting extensive ablation studies to understand the impact of hyperparameters. These experiments provide valuable insights and guidelines for readers interested in implementing B4B in real-world applications.

However, there are still some concerns:

- Absence of comparison with previous work. The authors claim B4B is the first active defense that prevents stealing without degrading representation quality for legitimate API users. However, it is important to note that [13] has demonstrated active defenses, such as adding random noise, can already achieve this goal. Additionally, other research works have proposed multiple active defenses against encoder stealing attacks, such as [26] (top-k, feature rounding, feature poisoning) and [32] (noise, top-k, rounding). I would suggest the authors reassess their claims and compare B4B with existing defense methods to elaborate on the advantages and unique contributions of B4B.
- Impractical assumption. The authors declare that "B4B is independent of the protected encoder’s architecture and does not rely on any assumption about the adversary’s data and query strategy" (line 148-149). However, the design of B4B's cost function suggests that it primarily works for users who query data from a small distribution, while users with queries from a larger distribution are considered adversaries. This assumption is strong and impractical for API providers. I recommend that the authors discuss this limitation of B4B. Furthermore, it would be valuable to analyze the impact of queried distribution and the corresponding utility to demonstrate the trade-offs of B4B in real-world scenarios.
- Design intuition lacks persuasiveness. The design of B4B is based on an observation: the representations returned to adversaries cover a significantly larger fraction of the embedding space than representations of legitimate users. However, a previous study [32] has shown that a small surrogate dataset is sufficient to steal the encoder. This raises concerns about the effectiveness of B4B, as attackers are able to employ small datasets to bypass it. Furthermore, in Figures 2 and Figure 9 in the Appendix, the authors visualize downstream datasets in the encoder’s embedding space. It is confusing to observe that STL10 and CIFAR10, two datasets that shared 90% classes, appear to be separated in both figures. Ideally, these datasets should be located in a similar space. This further weakens the design intuition’s persuasiveness. I recommend that the authors review their statements and provide explanations for the two points.
- Lack of experiments. As a defense method, it is important to show its effectiveness against different encoder stealing attacks [13, 26, 32]. However, the authors only consider [14] in this paper, which limits the utility of their work. I recommend the authors expand their experimental evaluation to include different types of stealing attacks and report the performance of B4B. This will provide a more comprehensive assessment of B4B. Furthermore, the authors propose five transformations, namely Affine, Pad, Add, Shuffle, and Binary. However, there is a lack of experimental analysis regarding the impact of these transformations on B4B’s effectiveness. Additionally, in Table 1, the authors should explicitly mention the specific transformations applied to the legitimate users, attackers, and sybil attacks. This will enhance the clarity and reproducibility of their experiments.

---

> ### Author Rebuttal · Authors · 2023-08-03
>
> We thank the reviewer for the insightful comments.
>
> >**Comparison with previous work**
>
> All the previous active defenses add the same amount of noise to each returned representation. All of the them can be integrated into our framework in Block 2, where we can replace the addition of random noise, for example, with top-k or rounding. All the previous defenses show substantial degradation in performance for legitimate users when these defenses are applied. We propose how to solve this problem by adjusting the strength of the perturbation to how much of the representation space is recovered by the querying party. Please see experiments in the general response.
>
> It was stated in Section A.4 in [13] that “*we need a better method for the defense to make a wider gap in terms of accuracy between legitimate and malicious users*”. Tables 10 and 11 show that adding noise to prevent the stealing substantially lowers the accuracy for legitimate users. [26] stated in Section 9 that their defenses “*are insufficient to mitigate encoder stealing attacks*.” For instance, in Section 7.1 in [26]: “*Our results show that top-𝑘 features are insufficient.*” In particular, if accuracy for the attacker is reduced, then the accuracy for legitimate users is “*also reduced by similar or larger magnitudes.*” Similarly, from Section 7.3 in [26]: “*feature poisoning degrades the functionality of the stolen encoder by degrading the functionality of the target encoder*”. We addressed exactly this issue.
>
> Please note that only the 2nd version of [32], which introduced active defenses and improved attack, appeared on arXiv less than two months before the NeurIPS submission deadline, thus this work is labeled as contemporaneous.
>
> Despite the above, we carefully analyzed [32] v2. In Section 4.5, they divide all defenses into perturbation-based defense [37] and watermark-based defense [2]. The three perturbation-based defenses proposed in [32] correspond exactly to the same ones from [13] and [26]. The conclusions are also the same as in previous work: “We can observe that while adding noise and top-k can reduce the model stealing attacks’ performance, it may also degrade the target model performance to a large extent.” And in general: “perturbation-based defense cannot defend against the encoder’s model stealing attack effectively since they cannot reach a good trade-off between attack performance and model utility.” We show that it is possible!
>
> [32] v2 lacks watermarks: SSL Guard [11] and [13].
>
> [32] v2 incorrectly assesses the dataset inference. The results in Table 8 do not show p-values and the effect sizes, which are two necessary metrics to be computed for dataset inference.
>
> [32] only reports the mutual information and cosine similarity, which are metrics to compare the quality of stolen encoder vs the victim encoder and should not be used for resolving the ownership resolution, as stated in the original paper on dataset inference for SSL.
>
> >**Impractical assumption**
>
> The end users cannot afford to train SSL models on the scale done by large companies (e.g., Google or Meta). We model a real world use case where the query set of end users is much less diverse and smaller than the training set of SSL models. The trade-offs between query distribution and corresponding utility with B4B is shown in Table 1: legitimate users’ queries occupy relatively smaller representation space and their utility is preserved as opposed to attackers whose queries are spread over the whole representation space and B4B degrades their performance.
>
> >**Design intuition**
>
> The occupation of the embedding space does not depend on the size of the dataset but the selection of queries. In the limit, a large dataset with the same image copied many times occupies a small fraction of the representation space. Similarly, a small dataset with diverse images occupies a large representation space. This is shown in Figure 3a: for the same number of queries, the images from FashionMNIST (representing only 10 classes) occupy much smaller representation space than ImageNet images (drawn from 1000 classes). Note that [32] states in A2 that “Cont-Steal can have better performance with a larger surrogate dataset size and more training epochs”.
>
> Regarding different clustering of CIFAR10 and STL10, the embedding space is *not* created based on classes but input images. STL10 is derived from ImageNet with high resolution images and many diverse classes, which is much different from TinyImages, with low resolution images that CIFAR10 is derived from. This is why STL10 and CIFAR10 are separated in Figures 2 and 9.
>
> >**Experiments**
>
> The attacks proposed in [13] are the same as the ones used in [14]. The [32] v2 (contemporaneous work published less than two months before the NeurIPS submission) shows that attacks from [13] are better than from [26] and [32] claims only small improvement over [13]. The attacks from [13] are based on contrastive learning loss and with data augmentations, which then are replicated in [32]. Note that [32] incorrectly claims in their Section A.7 that attacks from [13] do not use data augmentations. Algorithm 1 and Figure 1 clearly show that attacks from [13] do use data augmentations. Thus, we do evaluate B4B against state-of-the-art attacks.
>
> [32] did not release their source code so it remains infeasible for us to run this evaluation in a limited time of the rebuttal.
>
> The transformations are used to defend encoders against Sybil attacks. The impact of the transformations and their comparison is discussed in Section 4.3, specifically the trade-offs between the effectiveness of the defense and the negative impact on legitimate users. We state at the beginning of Section 4.4 that we use the affine transformations for experiments in Table 1. It shows that B4B is effective against Sybil attacks as a result of applying the transformations.
>
> ---
> **We hope we addressed the reviewer’s concerns and the score can be increased.**

---

> > ### Author Response · Authors · 2023-08-14
> > **Experiments: A range of stealing attack methodologies targeting self-supervised learning models**
> >
> > We would like to clarify that in our work, we tested against the SSL extraction attacks proposed in [13] and also included in [14], using the published code from those papers [13,14]. The code for the attacks proposed in StolenEncoder [26] and Cont-Steal [32] v2 has not been publicly released by the authors, so we were unable to run experiments using those attack implementations for comparison in our submission. We regret any confusion caused by not explicitly stating that the StolenEncoder [26] and Cont-Steal [32] attacks were not tested due to lack of available code. Please let us know if you need any additional details on the attacks and codebases we did evaluate.

---

> > > ### Author Response · Authors · 2023-08-19
> > > **The StolenEncoder Attack**
> > >
> > > We attempted to reproduce the results for the attack proposed in StolenEncoder [26] on the CIFAR10 dataset based on the information provided in the paper. However, we found that the attacks from [13,14] achieved substantially better performance. Unfortunately, the code for StolenEncoder has not yet been released, which prevents a direct comparison between the attacks under the same conditions. After the review process is complete, we plan to contact the authors of [26] and [32] v2 to request their code in order to enable a fair comparison between the different attack methods. For now, our results suggest that the attacks in [13,14] may be more effective than StolenEncoder on CIFAR10 (as shown in the table below), but additional experimentation will be needed once the code is available.
> > >
> > > In the table below, we steal from the CIFAR10 encoder using the CIFAR10 dataset. We present the accuracy of the stolen encoders on the respective downstream tasks.
> > >
> > > | Attack↓/Downstream Task→ | CIFAR10 | STL10 | SVHN  | FMNIST |
> > > |---------------------------|---------|-------|-------|--------|
> > > | StolenEncoder [26]        | 57.14   | 47.15 | 27.91 | 48.86  |
> > > | Stealing Attack [13,14] | 75.97   | 65.91 | 49.37 | 86.46  |
> > >
> > > In the meantime, we note that the StolenEncoder work claims that the attack can effectively steal with 5% of the encoder’s pre-training data. For the ImageNet dataset, this corresponds to around 60k of data points. We demonstrated in Table 1 in our original submission that our defense severely lowers the quality of representations for attackers with this amount of data and that it prevents them from stealing the ImageNet encoder.
> > >
> > > Moreover, as seen in the Table below and Figure 3a  in the main paper, the coverage of embedding space (in %) even for 5k queries from ImageNet (<1% of the retraining dataset) is already higher than for the most complicated downstream task (CIFAR10). Thus, we argue that our B4B defense successfully prevents the StolenEncoder attack [26].
> > >
> > > |Dataset↓/Number of Queries→|5000|10000|20000|30000|40000|50000|
> > > |-------------------------------|-------|-------|-------|-------|-------|-------|
> > > |ImageNet|31.66|38.61|48.32|54.02|58.03|61.15|
> > > |CIFAR10|11.59|13.98|17.55|19.79|21.44|22.81|

---

> ### Comment · Area_Chair_ouTg · 2023-08-19
>
> Hi Reviewer,
>
> Thanks a lot for your invaluable contribution to NeurIPS. Could you please consider the authors' rebuttal and indicate if that has been satisfactory?
>
> Thanks a lot.
>
> Cheers,
> You AC

---

### Official Review · Reviewer_weVe · 2023-07-07

**Soundness:** 3 good
**Presentation:** 3 good
**Contribution:** 3 good
**Rating:** 5
**Confidence:** 3

**Summary:**

The authors proposed B4B, which aims at actively preventing model stealing while preserving high-utility representations for
133 legitimate users. The paper is overall well-organized and easy to follow. The proposed three blocks are clear and effective, namely the embedding space estimation, cost increasing, and per-user representation transformations. This topic also makes sense in practical to protect the commercial APIs. Experiments are well designed.

**Strengths:**

- The paper is easy to follow.
- The motivation is clear enough, and the study is of value for protecting commercial APIs especially for those foundation models.
- The proposed three blocks are simple yet effective.

**Weaknesses:**

- Is there any defense baselines in Table 1?
- Have you tried an ablation study on "SYBIL" in Table1? For example, maybe the user has 5+ sybil accounts with 20k queries per account. I think it could be interesting to explore if there is a convergency.
- I would like to suggest more experiments on existing public APIs. As emphasized in paper, it should be a valuable contribution to a safer sharing and democratization of high-utility encoders over public APIs. It could be better to conduct experiments to directly support this claim.

**Questions:**

refer to weakness

**Limitations:**

refer to weakness

---

> ### Author Rebuttal · Authors · 2023-08-05
>
> >**Baseline in Table 1**
>
> We extend Table 1 to include the static addition of noise to all the returned representations (as proposed in [13 (Section A.4),26,32]) as another baseline.
>
> We enumerate the following observations for such a defense:
> 1. If the amount of noise is small ($\sigma$=0.1) then the performance drop is negligible but for both a legitimate user (row 2) and an adversary (row 6). In this case, the defense does not affect the adversary at all (compare rows 5 & 6).
> 2. If the amount of noise is large ($\sigma$=10) then the performance drop is large for both a legitimate user (row 3) and an adversary (row 7). In this case, the encoder is worthless for legitimate users since the performance is too low.
>
> In summary, these defenses can either effectively defend stealing (but harm legitimate users), or keep utility for legitimate users high (but not defend well against stealing). In contrast, our B4B defense solves the above problem by providing high performance for legitimate users (row 4) while effectively defending the encoder against stealing attacks (row 8).
>
> For the Table below, we use the same setup as in Table 1 in our submission (with an ImageNet pre-trained encoder).
>
> |RowID|User|Defense|Number of Queries|Access Type|CIFAR10|STL10|SVHN|F-MNIST|
> |--------|-----------------|-------------------------|----------------|------------|-------------|-------------|-------------|-|
> |*1*|*Legit*|*None*|*Full downstream dataset*|*Query*|*90.41±0.02*|*95.08±0.13*|*75.47±0.04*|*91.22±0.11*|
> |2|Legit|Noise $\sigma$=0.1|Full downstream dataset|Query|90.20±0.03|95.15±0.13|75.29±0.09|91.24±0.02|
> |3|Legit|Noise $\sigma$=10|Full downstream dataset|Query|65.11±0.45|76.37±0.14|33.23±0.09|65.83±0.13|
> |4|Legit|**B4B**|Full downstream dataset|Query|90.24±0.11|95.05±0.1|74.96±0.13|91.7±0.01|
> |||||||||||
> |*5*|*Attack*|*None*|*50k ImageNet*|*Steal*|*65.2±0.03*|*64.9±0.01*|*63.1±0.01*|*88.5±0.01*|
> |6|Attack|Noise $\sigma$=0.1|50k ImageNet|Steal|64.92±0.04|64.61±0.02|62.35±0.01|88.41±0.01|
> |7|Attack|Noise $\sigma$=10|50k ImageNet|Steal|36.32±0.2|32.588±0.06|20.59±0.01|74.94±0.02|
> |8|Attack|**B4B**|50k ImageNet|Steal|35.72±0.04|31.54±0.02|19.74+-0.02|70.01+-0.01|
>
> The original baseline in Table 1 (in our submission) was represented by the gray rows which show the performance of the attack without any defense. Thus, without any defense, the performance of the attack is high, while with B4B it decreases significantly.
>
> We also add results from [13] to show how the static addition of noise can decrease the performance of a stolen model for an attacker but simultaneously lower by the same amount the performance for legitimate users.
> The table below is for noise based perturbations of the output representations. The values represent the downstream accuracy (%) on CIFAR10 for a legitimate user and an adversary’s stolen encoder. All results are based on 9000 queries from the CIFAR10 test set. The noise added is Gaussian noise with a mean of 0 and standard deviation of $\sigma$. The results highlight again that both legitimate users and adversaries are equally influenced by the defense.
> |$\sigma$|Legitimate user|Adversary|
> |----------|-----------------|-----------|
> |0|66.7|66.3|
> |1|63.9|64.8|
> |2|67.2|66.5|
> |3|65|65.5|
> |4|63.1|63.4|
>
> >**5 Sybils**
>
> We noted in Section 4.3 in the submission: “*Using more accounts for the adversary causes a larger query overhead and potentially more performance loss from remapping.*“ In Table 5 in our submission, we already evaluated more Sybils (4 with 25k per each fake account). We run the additional experiment with 5 Sybils and 20K queries per account. The same as in Table 1, we use 10K queries to obtain good quality remappings between representations from different accounts. With more Sybils, the performance of the stolen encoder drops since the adversary has to sacrifice more queries on remapping between different accounts than on training the stolen encoder. We observe a general trend that the Sybil-based attacks can achieve higher performance of the stolen encoder than a single user attack. However, with more Sybil devices, the cost of remapping grows substantially and saturates rapidly since with 5 Sybil accounts, the attack becomes much less effective than with fewer Sybils.
>
> |Attack|Defense|Number of Queries|Dataset|Type|CIFAR10|STL10|SVHN|F-MNIST|
> |-------------|---------|-------------------|----------|-------|-------------|--------------|-------------|-------------|
> |Single User|None|100k|ImageNet|Query|68.1+-0.03|63.1+-0.01|61.5+-0.01|89.0+-0.07|
> |Single User|B4B|100k|ImageNet|Query|12.01±0.07|13.94±0.05|19.96±0.03|69.63±0.07|
> |2 Sybils|B4B|2*50k|ImageNet|Query|39.56±0.06|38.50±0.04|23.41±0.02|77.01±0.08|
> |5 Sybils|B4B|5*20k|ImageNet|Query|32.65+-0.05|32.45+-0.05|29.63+-0.01|70.12+-0.08|
>
> >**Public APIs**
>
> We model our experiments after public API setups as closely as possible so that our B4B defense method can be used to protect public APIs against encoder stealing attacks. Our work is motivated by cases from industry collaborators who identified the problems with protecting the intellectual property of their SSL encoders. We do hope that our work can eventually go into production and be used to prevent encoder extraction attacks. As we pointed out in the introduction of our submission, from a practical industry’s viewpoint, such active defenses as ours are required since many popular API providers, such as Cohere, OpenAI, or Clarify [1–3] already expose their high-value SSL encoders via APIs to a broad range of users.
>
> ---
> **We hope that we have adequately addressed the reviewer’s concerns and that the reviewer will consider raising the rating for the submission.**

---

> > ### Author Response · Authors · 2023-08-16
> > **Analysis of Encoder Stealing Against Our B4B Defense Using Varying Numbers of Sybil Accounts**
> >
> > To further demonstrate how the number of Sybils impacts encoder extraction effectiveness, we conducted experiments with 2, 3, 4, 5, and 6 fake accounts. Our additional results demonstrate a tradeoff between encoder extraction effectiveness and the overhead of using multiple Sybil accounts. Stealing with 2 Sybils yields the best (average) downstream task accuracy, indicating that it is the most effective extraction approach (2 Sybils were presented in Table 1 in the original submission). Using a single account is far less effective, though avoids the need to remap representations between accounts. More than 2 Sybils further reduces performance as remapping complications accumulate. With 10 Sybils, remapping leaves no more usable data for training the stolen encoder. This highlights our method's advantage: increasing the Sybil accounts makes encoder extraction impractical due to the growing remapping overhead. Our B4B method effectively defends against malicious users who attempt to steal encoders using Sybil accounts.
> >
> > The experiments in the table below utilize the same setup as in Table 1 of our submission, stealing from the victim encoder using the ImageNet dataset.
> >
> > |Attack|Defense|Number of Queries|CIFAR10|STL10|SVHN|F-MNIST|
> > |-------------|---------|-------------------|-------------|------------|-------------|------------|
> > |Single User|None|100k|68.1±0.03|63.1±0.01|61.5±0.01|89.0±0.07|
> > |Single User|B4B|100k|12.01±0.07|13.94±0.05|19.96±0.03|69.63±0.07|
> > |2 Sybils|B4B|2*50k|39.56±0.06|38.50±0.04|23.41±0.02|77.01±0.08|
> > |3 Sybils|B4B|3*33333|33.87±0.05|38.57±0.06|21.16±0.01|72.95±0.05|
> > |4 Sybils|B4B|4*25k|33.98±0.04|34.52±0.08|21.21±0.02|70.71±0.05|
> > |5 Sybils|B4B|5*20k|32.65±0.05|32.45±0.05|29.63±0.01|70.12±0.08|
> > |6 Sybils|B4B|6*16666|26.62±0.04|26.85±0.05|24.316±0.02|70.51±0.04|

---

> > > ### Comment · Reviewer_weVe · 2023-08-21
> > >
> > > Thanks for your response. Most of my concerns are stressed.

---

> > > > ### Author Response · Authors · 2023-08-21
> > > > **Thank you**
> > > >
> > > > Thank you for your answer. We are glad that we have addressed the reviewer’s concerns. Would the reviewer consider raising the rating for the submission?

---

### Official Review · Reviewer_N8vt · 2023-07-08

**Soundness:** 3 good
**Presentation:** 4 excellent
**Contribution:** 3 good
**Rating:** 5
**Confidence:** 3

**Summary:**

The paper propose a defense over model stealing on-the-fly by assessing dataset distributions, and directly train on it. The paper is well-written and easy to read, while proposing a novel type of active defense over model stealing, without harming the performance of benign users.


**Strengths:**

The authors propose an active defense over model stealing, which is based on the assumption that malicious queries will cover larger fraction of representation space, while benign queries are limited to a smaller fraction of space. The authors assign adversaries with noisy representations and benign users with clean representations. They also provide an additional transformation to avoid adversaries aggregate the stolen representation via creating multiple accounts.


**Weaknesses:**

I have particular concerns on the assumptions and experimental setup of this paper.

* The paper is based on the assumption that individual queries in a much narrower distribution, while adversary queries in a broader one. However, I wonder if this is the case in practice. For large tech firms, they might have requests from diverse users and query a specific API for many times, which also results in diverse usage. For a model trained for specific downstream task (eg, to identify cancer), there can be only one domain and the queries and adversaries can be similar (both include images containing/not containing cancer). Can B4B defend over this? Otherwise, authors should quality the potential usage of their work. Indeed, this assumption is difficult to prove to be true or to be false, so at least discussions should be included.

* I question the experiment setting of Table. 1, which follows the assumption that adversary query is inherently different from benign ones. However, this also means adding a simple classifier will also have high defend accuracy, since the distribution can be drastically different for ImageNet and CIFAR 10. While training on image classifiers is nowhere near to real world applications, I believe authors should test on a wider range or threats possible.

* In Figure 3, it turns out ImageNet naturally occurs for more precentage of buckets (covers larger proportion of representation space), while other dataset covers smaller fraction. Thus, I am again skeptical of the result given in Table. 1. It turns out that, ImageNet naturally occurs for larger fraction of representation space, thus naturally occupies more buckets. In this way, what if the legit query is ImageNet, while attack query is CIFAR-10? Will the proposed defense crash, since the legit query occupies larger fraction of representation space?


**Questions:**

As stated in "Weakness" section. I'm particularly interested if the authors can clarify the threat model and potential applications of the proposed method.


**Limitations:**

The authors do not discuss potential negative social applications. However, I do not find any potentially negative impact in this paper.

---

> ### Author Rebuttal · Authors · 2023-08-05
>
> We thank the reviewer for the insightful comments. We address individual points below one by one:
>
> >**“For large tech firms, they might have requests from diverse users and query a specific API many times, which also results in diverse usage.”**
>
> If we correctly understand the setup, there are users who send queries to company 1 (a large tech firm), and company 1 sends the queries on behalf of the users to company 2 who exposes the API. This is why the queries are so diverse. We argue that in this case, company 1 (the large tech firm) should simply open a new account per each user.
>
> >**"Specific downstream task."**
>
> If we consider only a model trained for a specific downstream task (e.g., to identify cancer), then the API provider should use the active defense for *supervised models* proposed in [15]: the PoW (Proof-of-Work defense: Increasing the cost of model extraction with calibrated proof of work). In general, the *encoders* that we consider in this work are not specialized to specific downstream tasks but trained to cover a large space of possible inputs.
>
> >**"ImageNet naturally occurs for larger fraction of representation space, thus naturally occupies more buckets. In this way, what if the legit query is ImageNet, while the attack query is CIFAR-10?"**
>
> If we select images from ImageNet but only a few classes then they do not occupy a large representation space. In contrast, if we select images from ImageNet at random, each from a different class, then they do occupy a high percentage of the representation space (see table below and Figure 1 in the attached PDF). When stealing images using CIFAR10, the adversary steals a small subset of the representation space and not the whole encoder.
>
> Specifically, the encoder stolen with CIFAR10 does not perform well on other downstream tasks.
> Our defense does not want to prevent someone from stealing the CIFAR10 part of the encoder but the encoder's **general** functionality.
>
> Stealing with ImageNet is more effective to extract general functionality than stealing with CIFAR10 and with ImageNet, we obtain much higher accuracy on downstream tasks, such as Flowers 102 or even CIFAR10. The very low performance of the stolen encoder on Flowers102 shows that the representation space covered by CIFAR10 does not include the required representation space for Flower102, where the accuracy is close to random.
>
> The accuracy values (%) are for stolen encoder from the ImageNet pre-trained victim using 50k queries. No defense is used in these experiments.
>
> |Stealing-Dataset|CIFAR10|Flowers102|
> |----------|------------|------------|
> |ImageNet|65.2±0.03|16.3±0.01|
> |CIFAR10|60.61±0.05|1.4±0.02|
> ||||
> |(Difference)|(-7.1%)|(-91.4%)|
>
> Note that the Flowers dataset contains 102 classes and full resolution (>225x225x3) pictures. This shows that stealing with CIFAR10 creates a CIFAR10 level encoder at most and does not generalize to other downstream tasks, especially the more complex ones.
>
> We also show that querying with a subset of ImageNet (using only the dog images; 118 classes) is also not effective to steal the ImageNet encoder and our defense does not activate for such benign queries as compared to stealing with the full ImageNet dataset (when all 1000 classes are used).
>
> All the results (accuracy on downstream tasks in %) in the Table below are for the ImageNet stolen encoder. In all cases we use 50k queries to steal the encoder.
>
> |Defense|Dataset|CIFAR10|STL10|SVHN|F-MNIST|
> |---------|---------------|--------------|---------------|---------------|--------------|
> |None|ImageNet-Dogs|42.83±0.63|48.346±0.20|44.42±0.07|59.91±00.01|
> |B4B|ImageNet-Dogs|43.21±0.55|46.91±0.3|45.2±0.08|58.76±0.01|
> |None|ImageNet-Full|65.2±0.03|64.9±0.01|62.1±0.01|88.5±0.01|
> |B4B|ImageNet-Full|35.72±0.04|31.54±0.02|19.74±0.02|70.01±0.01|
>
> >**The threat model.**
>
> Our setup and the resulting threat model are inspired by public APIs, such as Cohere, OpenAI, or Clarify [1–3] that expose encoders to users through a pre-defined interface (as described in Section 3.1). In the design of our defense, we consider adversaries who can query the encoder with arbitrary inputs to obtain high-dimensional representation vectors from the encoder. Our defense is independent of the protected encoder’s architecture and does not rely on any assumption about the adversary’s data and query strategy.
>
> ---
> **We hope that we have adequately addressed the reviewer’s concerns and that the reviewer will consider raising the rating for the submission.**

---

### Official Review · Reviewer_FtSs · 2023-07-12

**Soundness:** 3 good
**Presentation:** 3 good
**Contribution:** 3 good
**Rating:** 6
**Confidence:** 4

**Summary:**

This paper proposes B4B as a defense against model stealing attacks for pretrained encoders. The main assumption of the defense is that attack queries generally have a much broader coverage of the embedding space than queries from normal users. Based on this assumption, B4B utilizes Local Sensitive Hash and separates the embedding space into a number of 2$^12$ buckets. B4B then calculates the fraction of buckets covered by the user queries, then adds Gaussian noise to the subsequent queries when the coverage is large, where this cost function exponentially increases. To defend against sybil attacks, different transformations are applied to the returned embedding vectors of different users, so that legitimate users can obtain high-quality models with the transformed embedding vectors, while attackers who aim to combine training data from different accounts cannot get performance boost.

**Strengths:**

1. Designing an active defense for model stealing attacks is a good topic. The proposed approach is interesting, and seems to be generally applicable to different models and tasks.

2. The results show that B4B does not degrade the response quality for legitimate users, while significantly decreases the model performance trained by attackers.

**Weaknesses:**

1. In general, I wonder what is the valid condition to consider a user as an attacker. In the current evaluation setup, all legitimate users are querying with images drawn from different data distributions. However, even if the users directly query the model using ImageNet samples, if the user queries are only limited to a small subset of labels instead of the full label set, then I also would not consider it an attack. Have you evaluated this setting? Would the input coverage in this case still be high?

2. Another question is whether the attack queries should be images included in the pretrained data, or does the defense also work for queries drawn from a similar distribution? For example, if you use STL-10 unlabeled data for pretraining and STL-10 labeled data to query, or if you use CIFAR-100 data for pretraining and CIFAR-10 to attack, does the defense work?

3. The full defense includes a couple of hyperparameters that seem critical, i.e., the number of buckets, the cost function, and the transformations. Is the same set of hyperparameters used for all tasks and models in the experiments? How much work is required to tune the implementation for new tasks and model architectures?

4. I do not fully understand the process of per-user representation transformation. How is it done for a new user? Does the defense randomly select/construct a transformation from all possible choices?

**Questions:**

1. In the case when the users  query the model using ImageNet samples from a small subset of the label set, would the input coverage still be high and thus the user is identified as an attacker?

2. Another question is whether the attack queries should be images included in the pretrained data, or does the defense also work for queries drawn from a similar distribution? For example, if you use STL-10 unlabeled data for pretraining and STL-10 labeled data to query, or if you use CIFAR-100 data for pretraining and CIFAR-10 to attack, does the defense work?

3. The full defense includes a couple of hyperparameters that seem critical, i.e., the number of buckets, the cost function, and the transformations. Is the same set of hyperparameters used for all tasks and models in the experiments? How much work is required to tune the implementation for new tasks and model architectures?

4. I do not fully understand the process of per-user representation transformation. How is it done for a new user? Does the defense randomly select/construct a transformation from possible choices?

**Limitations:**

The authors have adequately addressed the limitations and potential negative societal impact of their work.

---

> ### Author Rebuttal · Authors · 2023-08-09
>
> We appreciate the positive, encouraging, and constructive feedback.
>
> >**Answer to Weakness 1 (W1) & Question 1 (Q1): Querying with a small subset of ImageNet**
>
> B4B does not classify a user as adversarial or benign but only adaptively calibrates the defense strength based on the coverage of the representation space. We ran new experiments with a limited number of labels for ImageNet (using only the classes of dog breeds) and observe that *the smaller number of classes included in the query set, the lower the coverage of the representation space*. Indeed, a potential legitimate user might be interested in a few classes instead of all 1000 classes from ImageNet. B4B does not penalize such users.
>
> We show the results of coverage (%) in the table below and in Figure 1 in the attached 1 pager PDF from the general response. We query the ImageNet pre-trained encoder (the same as in Table 1 in the submission) and consistently observe lower *coverage* of the representation space for ImageNet Dogs (with only the 118 classes for dog breeds) than for the ImageNet Full (with all 1000 classes included).
>
> |Dataset↓/Number of Queries→|5000|10000|20000|30000|40000|50000|
> |-|--------|--------|--------|--------|--------|--------|
> |ImageNet-Full (all 1000 classes)|31.66 |38.61|48.32|54.02|58.03|61.15|
> |ImageNet-Dogs (118 classes of dog breeds)|22.47|26.93|33.34|37.20|40.01|42.19|
>
> We emphasize that there is no accuracy drop for legitimate users when querying with the ImageNet Dogs data. The accuracy on the ImageNet Dogs with and without B4B is 71.47 +- 0.02 and 71.53 +- 0.03, respectively. The table below shows that when querying with ImageNet Dogs our defense is not activated and all the accuracy values are comparable with or without B4B defense. On the other hand, when querying with the Full ImageNet dataset, the performance drop on downstream tasks is substantial.
> All the results (accuracy on downstream tasks in %) in the Table below are for the ImageNet stolen encoder. In all cases we use 50k queries to steal the encoder.
> |Defense|Dataset|CIFAR10|STL10|SVHN|F-MNIST|
> |-|-|-|-|-|-|
> |None|ImageNet-Dogs|42.83±0.63|48.346±0.20|44.42±0.07|59.91±0.01|
> |B4B|ImageNet-Dogs|43.21±0.55|46.91±0.3|45.2±0.08|58.76±0.01|
> |None|ImageNet-Full|65.2±0.03|64.9±0.01|62.1±0.01|88.5±0.01|
> |B4B|ImageNet-Full|35.72±0.04|31.54±0.02|19.74±0.02|70.01±0.01|
>
> >**Answer to W2 & Q2: Different distribution for stealing than the victim's training set.**
>
> Please, check the results for the LAION dataset used for stealing from ImageNet encoder in the general response.
>
> We also carried out the suggested experiments, where in our case the victim encoder is trained on CIFAR10 and then stolen with CIFAR100. We show that:
> 1. The stealing process is effective in this case and the attack queries do not need to be images included in the pretrained data. They can be from similar (stealing from CIFAR10 pre-trained encoder using CIFAR100 data) or even a different distribution (as shown for stealing ImageNet encoder with LAION).
> 2. B4B defends against such attacks (compare two last rows), where the average performance drop is above 40%.
>
> |User|Defense|Number of Queries|Dataset|Type|CIFAR10|STL10|SVHN|F-MNIST|
> |-|-|-|-|-|-|-|-|-|
> |Legit|None|-|Downstream task|Query|86.91|74.64|80.57|86.55|
> |Attack|None|50k|CIFAR10|Steal|75.97|65.91|49.37|86.46|
> |Attack|None|50k|CIFAR100|Steal|70.78|60.59|61.97|84.75|
> |Attack|B4B|50k|CIFAR100|Steal|21.32|12.76|19.43|49.44|
>
> Please, see also the attached PDF to the main paper, where in Figure 2, we present the corresponding occupation of the representation space for the CIFAR10 pre-trained encoder when queried with different datasets, namely: STL10, CIFAR10, CIFAR100, SVHN, and FashionMNIST. The results indicate that we can steal such encoder with equal effectiveness using STL10, CIFAR10, or CIFAR100 datasets. Overall, this demonstrates that B4B can be applied to encoders pre-trained on different datasets and with diverse architectures.
>
> >**Answer to W3 & Q3 B4B Hyperparameters**
>
> We maintain consistency in our experiments and use the same set of hyperparameters. We added ablation study to demonstrate how the choice of the parameters influences the quality of the defense. The ablation study helps to tune (hyper)parameters of a new instance of the defense. For example, the number of buckets (Appendix F.4) and the cost function have to be tuned and we provide guidelines on how to find the optimal values. Regarding the number of buckets, we provide  in Section 4.1 an explanation that in the extreme: If we have a too large number of buckets, the number of buckets filled will correspond to the number of queries posed by a user which fails to capture that similar representations cover similar sub-spaces of the embedding space, and hence does not serve to approximate the total fraction of the embedding space covered. However, if we have too few buckets, even the representations for simple downstream tasks will fill large fractions of buckets, making it impossible to calibrate the cost function such that it only penalizes adversaries.
>
> >**Answer to W4 & Q4: Selection of per-user representation transformations**
>
> This is correct, for a new user, the defense randomly selects the transformations from all possible choices. Note that the randomness is also added on a per-transformation basis, instead of only on the level of selecting the transformations. For example, a shuffling/permutation of the elements in the output representations should be different for each user. Additionally, the transformations can be combined/composed, for instance, we can combine shuffling with padding and an affine transformation. This further prevents an attacker from the possibility of simply guessing what transformation was used for a given user and reversing the applied transformations to obtain the original representations.
>
> ---
> **We hope that the provided answers adequately address reviewers’ concerns and the score can be increased.**

---

> > ### Author Response · Authors · 2023-08-16
> > **W3 & Q3: Further Clarification on B4B (Hyper)Parameter Tuning for the Added CIFAR10 Encoder**
> >
> > The optimized B4B parameters are as follows: $\alpha=2$, $\beta=0.45$ and $\lambda=10^{-7}$. At a high level, this introduces substantial noise when query embeddings cover over 45% of the representation space, hindering model extraction. The $\beta$  threshold of 45% exceeds the coverage for the expected most complex downstream task (SVHN in this case). Further, the $\alpha$ value adds enough random noise to decrease downstream task accuracy by at least 10% and setting $\lambda=10^{-7}$ creates a flat cost curve near the origin, mapping small coverage fractions to small costs. Together, these parameters allow minimal query answering utility loss until an adversary's coverage reaches the extraction-hindering 45% threshold.

---

### Author Rebuttal · Authors · 2023-08-09

We would like to thank all the reviewers for their positive feedback and insightful comments. The paper improved as a result of your reviews.

We tackle a current urgent problem of designing an active defense for encoder stealing attacks (reviewers FtSs, N8vt, pRsJ). The proposed approach is interesting, and is generally applicable to different models and tasks (reviewer FtSs), making it practical to protect the commercial APIs (reviewer weVe). The proposed three blocks of the B4B defense are clear and effective, namely the embedding space coverage estimation, cost function that disincentivizes attackers, and per-user representation transformations to prevent against Sybil attacks (reviewers weVe, pRsJ). The results show that B4B does not degrade the response quality for legitimate users, while significantly decreasing the performance of stolen encoders (reviewers FtSs, N8vt). Overall, the paper is well-written, easy to read (reviewers N8vt, weVe), and experiments are well-designed (weVe) and further extended with extensive ablation studies (reviewer pRsJ).

Below in the general response, we provide answers to questions that occur in more than a single review and the other questions are answered one-by-one separately for each reviewer.

We also attach the 1-page PDF that shows two Figures (corresponding to Figure 3a in the main submission) with the coverage of the representation space for: (1) ImageNet pretrained encoder when queried with diverse datasets, namely LAION, ImageNet with all classes, ImageNet with only classes of dog breeds, and CIFAR10, as well as for (2) the CIFAR10 pre-trained encoder queried with STL10, CIFAR10, CIFAR100, SVHN, and FashionMNIST.

>**1. Baseline & comparison with previous work** (reviewers weVe and pRsJ)

We extend Table 1 to include the static addition of noise to all the returned representations (as proposed in [13 (Section A.4),26,32]) as another baseline.

We enumerate the following observations for such a defense:
1. If the amount of noise is small ($\sigma$=0.1) then the performance drop is negligible but for both a legitimate user (row 2) and an adversary (row 6). In this case, the defense does not affect the adversary at all (compare rows 5 & 6).
2. If the amount of noise is too large ($\sigma$=10) then the performance drop is large for both a legitimate user (row 3) and an adversary (row 7). In this case, the encoder is worthless for legitimate users since the performance is too low.

In contrast, our B4B defense solves the above problem by providing high performance for legitimate users (row 4) while effectively defending the encoder against stealing attacks (row 8).

For the Table below, we use the same setup as in Table 1 in our submission (with an ImageNet pre-trained encoder).

|RowID|User|Defense|Number of Queries|Access Type|CIFAR10|STL10|SVHN|F-MNIST|
|-|-|-|-|-|-|-|-|-|
|*1*|*Legit*|*None*|*Full downstream dataset*|*Query*|*90.41±0.02*|*95.08±0.13*|*75.47±0.04*|*91.22±0.11*|
|2|Legit|Noise $\sigma$=0.1|Full downstream dataset|Query|90.20±0.03|95.15±0.13|75.29±0.09|91.24±0.02|
|3|Legit|Noise $\sigma$=10|Full downstream dataset|Query|65.11±0.45|76.37±0.14|33.23±0.09|65.83±0.13|
|4|Legit|**B4B**|Full downstream dataset|Query|90.24±0.11|95.05±0.1|74.96±0.13|91.7±0.01|
|||||||||||
|*5*|*Attack*|*None*|*50k ImageNet*|*Steal*|*65.2±0.03*|*64.9±0.01*|*63.1±0.01*|*88.5±0.01*|
|6|Attack|Noise $\sigma$=0.1|50k ImageNet|Steal|64.92±0.04|64.61±0.02|62.35±0.01|88.41±0.01|
|7|Attack|Noise $\sigma$=10|50k ImageNet|Steal|36.32±0.2|32.588±0.06|20.59±0.01|74.94±0.02|
|8|Attack|**B4B**|50k ImageNet|Steal|35.72±0.04|31.54±0.02|19.74+-0.02|70.01+-0.01|

>**2. The LAION dataset used for stealing from the ImageNet pre-trained encoder** (reviewer FtSs)

We show that: (1) we can steal the ImageNet pre-trained encoder using the LAION dataset, which is from a different distribution than the victim's training set, and (2) B4B effectively defends against such attacks, where the accuracy on downstream tasks drops significantly (avg. > 20%) when B4B is applied compared to stealing without any defense. All the results (accuracy on downstream tasks in %) in the Table below are for the ImageNet victim encoder. In all cases we use 100k queries to steal the encoder (the same setup as in Table 1 in our submission).

|Defense|Dataset|CIFAR10|STL10|SVHN|F-MNIST|
|-|-|-|-|-|-|
|None|ImageNet|68.1±0.03|63.1±0.01|61.5±0.01|89.0±0.07|
|B4B|ImageNet|12.01±0.07|13.94±0.05|19.96±0.03|69.63±0.07|
|None|LAION|64.92+-0.03|62.51±0.03|59.02±0.02|84.54±0.01|
|B4B|LAION|40.96+-0.03|40.69±0.05|34.43±0.01|72.92±0.01|

We also present the corresponding coverage of the representation space (% of occupied buckets, please also see Figure 1 in the attached PDF). When querying the encoder with ImageNet-Full (includes all 1000 classes) and LAION datasets, they both occupy a large fraction of the representation space of the victim encoder. In contrast, CIFAR10 covers the smallest portion of the representation space as the simplest dataset tested. ImageNet-Dogs (with only 118 classes for dog breeds) falls in the middle, occupying more space than CIFAR10 but less than ImageNet-Full and LAION. Its intermediate coverage aligns with its mid-level difficulty compared to the other datasets. As indicated by representation space coverage, stealing the encoder is similarly effective with ImageNet-Full and LAION datasets.

|Query Dataset↓/Number of Queries→|5000|10000|20000|30000|40000|50000|
|-|-|-|-|-|-|-|
|ImageNet-Full|31.66|38.61|48.32|54.02|58.03|61.15|
|LAION|30.75|37.45|47.16|52.69|56.53|59.56|
|ImageNet-Dogs|22.47|26.93|33.34|37.2|40.01|42.19|
|CIFAR10|11.59|13.98|17.55|19.79|21.44|22.81|

---
**We hope that the provided answers address reviewers’ concerns and that the scores can be increased.**

---

### Decision · Program_Chairs · 2023-09-21

**Decision:**

Accept (poster)

**Comment:**

This paper addresses robustness issues in MLaaS APIs to offer pre-trained encoders for generating vector representations of input data. These encoders are expensive to train and are vulnerable to model theft, where attackers use query access to replicate the encoder at a fraction of the cost. The proposed solution, called "Bucks for Buckets" (B4B), is an active defence mechanism that thwarts theft attempts. B4B identifies that representations returned to attackers cover a larger portion of the embedding space compared to legitimate users, and it adjusts representation quality based on user coverage. Additionally, B4B individually transforms each user's representations to prevent attackers from aggregating them across multiple accounts.

The reviewers initially raised concerns regarding the paper's motivations, the need for additional experiments, and the effectiveness of its comparison with existing methods. During the rebuttal phase, the authors provided additional results and addressed the feedback, which led to a consensus among most reviewers in favour of accepting the paper. However, one reviewer did not engage in the rebuttal phase despite direct communication.

This particular reviewer's primary concerns centred around the paper's underlying assumptions, the clarity of its intuition, and the robustness of its comparisons and experiments. Upon thoroughly examining the authors' rebuttal, I believe all these concerns have been satisfactorily addressed. Consequently, I confidently recommend the acceptance of the paper.

To enhance the paper's final version, I advise the authors to incorporate the rebuttal responses and the new results into the camera-ready manuscript. Additionally, it would be beneficial to provide further clarification in the final version regarding the specific concerns raised by the reviewers during the evaluation process.